# Cytoplasmic sequestration of p53 by lncRNA-CIRPILalleviates myocardial ischemia/reperfusion injury

Yuan Jiang [1,2,3,5], Ying Yang[1,5], Yang Zhang [1,5], Jiqin Yang[1], Man-man Zhang[1], Shangxuan Li[1], Genlong Xue[1], Xingda Li[1], Xiaofang Zhang [1], Jiming Yang[1], Xiang Huang[1], Qihe Huang[1], Hongli Shan [4], Yanjie Lu[1], Baofeng Yang [1,2✉] & Zhenwei Pan [1,2✉]

Myocardial ischemia/reperfusion (MI/R) injury is a pathological process that seriously affects the health of patients with coronary artery disease. Long non-coding RNAs (lncRNAs) represents a new class of regulators of diverse biological processes and disease conditions, the study aims to discover the pivotal lncRNA in MI/R injury. The microarray screening identifies a down-regulated heart-enriched lncRNA-CIRPIL (Cardiac ischemia reperfusion associated p53 interacting lncRNA, lncCIRPIL) from the hearts of I/R mice. LncCIRPIL inhibits apoptosis of cultured cardiomyocytes exposed to anoxia/reoxygenation (A/R). Cardiac-specific transgenic overexpression of lncCIRPIL alleviates I/R injury in mice, while knockout of lncCIRPIL exacerbates cardiac I/R injury. LncCIRPIL locates in the cytoplasm and physically interacts with p53, which leads to the cytoplasmic sequestration and the acceleration of ubiquitin-mediated degradation of p53 triggered by E3 ligases CHIP, COP1 and MDM2. p53 overexpression abrogates the protective effects of lncCIRPIL. Notably, the human fragment of conserved lncCIRPIL mimics the protective effects of the full-length lncCIRPIL on cultured human AC16 cells. Collectively, lncCIRPIL exerts its cardioprotective action via sequestering p53 in the cytoplasm and facilitating its ubiquitin-mediated degradation. The study highlights a unique mechanism in p53 signal pathway and broadens our understanding of the molecular mechanisms of MI/R injury.

[1] Department of Pharmacology (State-Province Key Laboratories of Biomedicine-Pharmaceutics of China, Key Laboratory of Cardiovascular Research, Ministry of Education), College of Pharmacy, Harbin Medical University, 150086 Harbin, Heilongjiang, P. R. China. [2] Research Unit of Noninfectious Chronic Diseases in Frigid Zone, Chinese Academy of Medical Sciences, 2019 Research Unit 070, 150086 Harbin, Heilongjiang, P. R. China. [3] Department of Cardiology, Sun Yat-sen Memorial Hospital, Sun Yat-sen University, 510120 Guangzhou, Guangdong, P. R. China. [4] Shanghai Frontiers Science Research Center for Druggability of Cardiovascular noncoding RNA, Institute for Frontier Medical Technology, Shanghai University of Engineering Science, 201620 Shanghai, China. [5] These authors contributed equally: Yuan Jiang, Ying Yang, Yang Zhang. ✉email: yangbf@ems.hrbmu.edu.cn; panzw@ems.hrbmu.edu.cn

Acute myocardial infarction (AMI) is a major cause of morbidity and mortality worldwide. Reperfusion therapies by coronary intervention such as primary percutaneous coronary intervention are the current standard strategy for AMI[1]. However, ischemia reperfusion (I/R) injury may occur after reperfusion and lead to worsening of myocardial damage[2]. To date, there remains a lack of effective treatments and preventative measures for I/R injury[3]. It is still urgent to deepen the understanding of the molecular mechanisms of I/R injury and to develop novel therapeutic interventions.

Long noncoding RNAs (lncRNAs) are a class of single strand non-protein coding transcripts that are more than 200 nucleotides (nt) in length[4]. LncRNAs exert their biological function with versatile action modes due to their complex interactome (DNA, RNA and protein) in the cell[4,5]. Tumor suppressor p53 functions as a master regulator of cell apoptosis[6]. Under normal condition, only small amount of p53 expresses and locates in the cytosol. Upon noxious stress, p53 is dramatically upregulated and translocates to the nucleus to trigger the transcription of apoptosis-related genes such as p53 upregulated modulator of apoptosis (Puma), Bcl2-associated X protein (Bax) and Noxa etc. to initiate cell apoptosis[7,8]. Apoptosis is a critical process in cardiac I/R injury[9]. Activation of p53 expression during I/R injury were shown to regulate cardiomyocyte apoptosis[10]. Interferon regulatory factor 9 was reported to promote ubiquitination degradation of p53 and protect heart from I/R injury by suppressing the transcription of Sirt1[11]. The degradation of p53 is mainly executed in the cytoplasm via ubiquitin-protease pathway mediated by a group of E3 ligases, such as murine double minute 2 (MDM2), constitutive photomorphogenic 1(COP1), and carboxy-terminus of Hsc70 interacting protein (CHIP) etc[12]. It remains unclear whether lncRNAs can regulate cardiac I/R injury via affecting the stability of p53.

Considering the large number and versatile action modes of lncRNAs, we hypothesized that there exists certain undiscovered lncRNA that critically controls cardiac I/R injury through modulating the stability of p53. In this study, we identified a novel lncRNA NONMMUT053812 by lncRNA microarray, which can inhibit cardiac I/R injury by fine-tuning cellular distribution and degradation of p53. For convenience, we named lncRNA-NONMMUT053812 as cardiac ischemia reperfusion associated p53 interacting lncRNA (lncCIRPIL).

## Results

### Downregulation of lncCIRPIL in I/R hearts and A/R NMCMs.
To study lncRNA transcriptome changes during cardiac I/R injury, we performed a lncRNA microarray analysis of ischemic zone (IZ), border zone (BZ), and non-ischemic zone (NIZ) of I/R hearts in mice. As shown in Fig. 1a, compared with non-ischemic zone, a total of 1976 lncRNAs were deregulated in ischemic zone, with 835 upregulated and 1141 down-regulated (P value < 0.05; Fold change > 1.2). In comparison with the border zone, 897 lncRNAs were altered in ischemic zone, with 489 upregulated and 407 down-regulated (P < 0.05; Fold change > 1.2) (Fig. 1a, b). Among deregulated lncRNAs, 563 of lncRNAs showed the same change in ischemic zone as compared with border zone or non-ischemic zone, with 256 upregulated and 307 down-regulated. When the fold change is set to >1.5, 96 lncRNAs showed consistent expression changes in ischemic zone, with 56 upregulated and 40 down-regulated. Among them, 31 were intergenic lncRNAs (Fig. 1a, b). The filtering method was shown in Fig. 1c. Studies showed that tissue-specificity of lncRNAs reflects their function[13–15]. Of these 31 intergenic lncRNAs, 19 were heart-enriched, so we performed the qRT-PCR assay to verified their expression level in I/R heart tissue. The qRT-PCR assay verified

that 3 intergenic lncRNAs were upregulated, and 11 down-regulated in I/R hearts of mice than sham controls (Fig. 1d). Of these 14 intergenic lncRNAs, heart-enriched NONMMUT053812 which locates on chromosome 5 with a length of 776 nts had a significant change after I/R injury. CPC2 is a fast and accurate online coding potential calculator based on sequence intrinsic features[16], its bioinformatic analysis identifies a set of four intrinsic features as Fickett TESTCODE score, open reading frame (ORF) length, ORF integrity and isoelectric point (pI), the analysis results shown in Supplementary Table 1 suggested that NONMMUT053812 possesses no protein coding potential. For convenience, NONMMUT053812 was named as cardiac ischemia reperfusion associated p53 interacting lncRNA (lncCIRPIL) and focused on for further study.

We firstly validated the expression change of lncCIRPIL in an in vitro ischemia model. The data showed that lncCIRPIL was significantly decreased in cultured NMCMs exposed to A/R (Fig. 1e). The fluorescent in situ hybridization (FISH) assay showed that lncCIRPIL mainly distributed in cytoplasm of cultured NMCM and isolated adult mouse ventricular myocyte, which was decreased after exposing to A/R or I/R injury (Fig. 1f).

### LncCIRPIL prevents apoptosis of cardiomyocytes.
To understand the role of lncCIRPIL in cardiac I/R injury, we examined the influence of lncCIRPIL on NMCMs subjected to A/R treatment. Overexpression plasmid or shRNAs were transfected into NMCMs to induce overexpression or knockdown of LncCIRPIL (Fig. 2a, b; Supplementary Fig. 1a). The ratio of lactate dehydrogenase (LDH) level in the medium of cultured NMCMs was increased upon A/R insult. The change was inhibited by overexpression of lncCIRPIL, while further exacerbated by lncCIRPIL-shRNAs (Fig. 2c; Supplementary Fig. 1b). Subsequently, TUNEL staining and flow cytometry analysis corroborated that LncCIRPIL overexpression reduced, whereas knockdown of lncCIRPIL aggravated A/R-induced apoptosis of NMCMs (Fig. 2d; Supplementary Fig. 1c). Meanwhile, lncCIRPIL overexpression restored, while knockdown of lncCIRPIL further suppressed Bcl2/Bax ratio in NMCMs exposed to A/R (Fig. 2e; Supplementary Fig. 1d). Additionally, overexpression of LncCIRPIL inhibited, while knockdown of lncCIRPIL further aggravated the increase of caspase-3, caspase-8, caspase-9 activities of NMCMs exposed to A/R insult (Fig. 2f; Supplementary Fig. 1e). The data indicated that lncCIRPIL protects cardiomyocytes from A/R-induced injury by inhibiting apoptosis in vitro.

### Cardiac-specific transgenic overexpression of lncCIRPIL alleviated cardiac ischemia reperfusion injury.
To further confirm the protective role of lncCIRPIL against cardiac I/R injury, we generated cardiac-specific lncCIRPIL transgenic overexpressing mice (lncCIRPIL-TG) (Supplementary Fig. 2a). First, we confirmed in major organs of mice that LncCIRPIL transgene specific triggered the expression of LncCIRPIL in heart, even in cardiomyocytes (Fig. 3a; Supplementary Fig. 2b). Echocardiography study showed no significant difference in the function of hearts between WT and lncCIRPIL-TG mice at the age of 8 weeks. Upon cardiac I/R injury, overexpression of lncCIRPIL prominently preserved cardiac function, as manifested by the restoration of ejection fraction (EF) and fractional shortening (FS) (Fig. 3b). The Evans blue and triphenyl-tetrazolium chloride (TTC) staining revealed that lncCIRPIL-TG mice had marked reduction of infarct area (IA) to area at risk (AAR) ratio as compared to WT mice (Fig. 3c, d). Both serum creatine kinase (CK-MB) and LDH levels were increased after I/R injury, which were suppressed by transgenic overexpression of lncCIRPIL (Fig. 3e, f). Moreover, the caspase-3 activity and TUNEL positive rate were lower in hearts

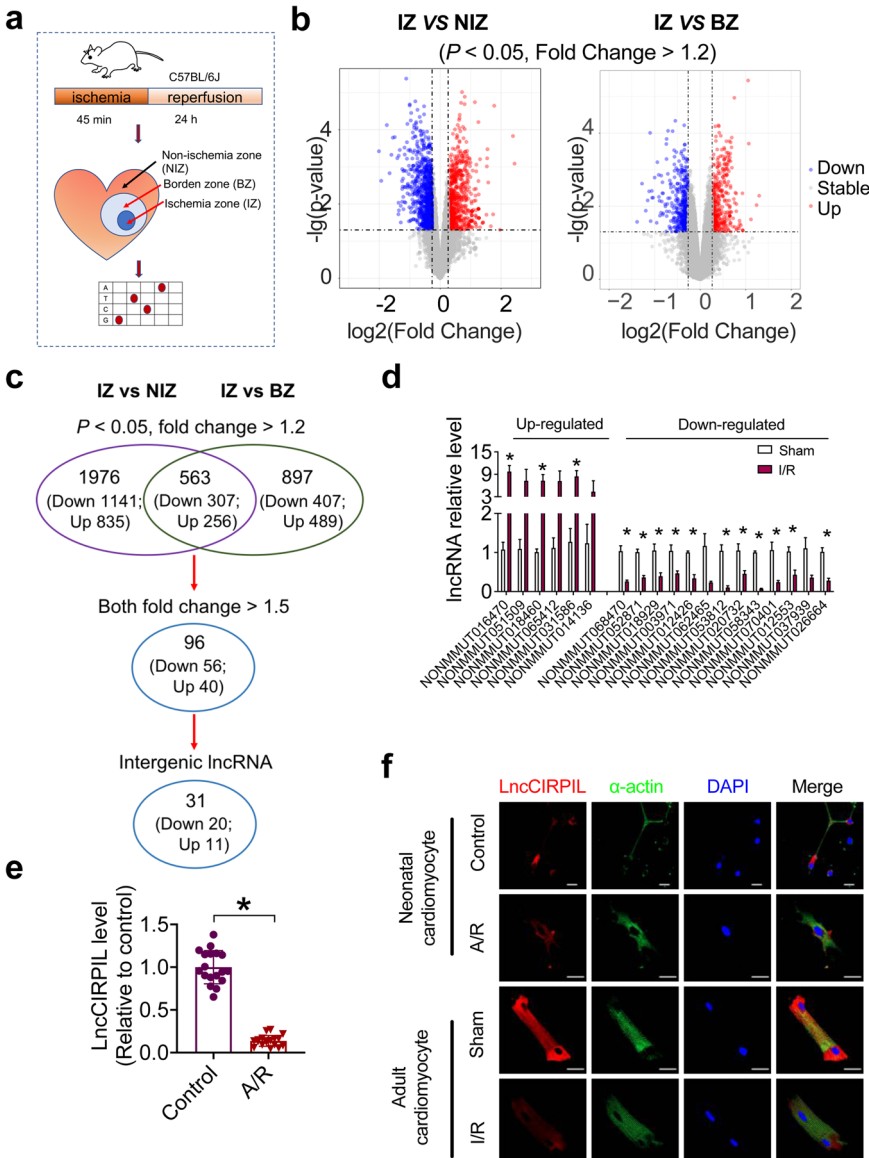

**Fig. 1 LncRNA-CIRPIL is downregulated in ischemia/reperfusion (I/R) hearts and anoxia/reoxygenation (A/R) NMCMs. a** The flowchart of study of I/R-induced expression profiles of murine lncRNA by MTA 1.0 transcriptome microarray assay. **b** The volcano plots of the I/R-induced expression profiles of murine lncRNAs in ischemia zone (IZ) compared with non-ischemia zone (NIZ) and border zone (BZ). $n = 3$. **c** Analysis of dysregulated lncRNAs in the hearts of I/R mice based on microarray profiling. **d** Validation of the heart-enriched and with more than 1.5 folds change intergenic lncRNAs in the hearts of I/R mice. $n = 5$. *$P < 0.05$ vs sham by two-tailed Student's $t$ test. **e** The expression changes of lncCIRPIL in NMCMs exposed to A/R. $n = 18$ samples from 6 independent experiments. *$P < 0.05$ vs control by two-tailed Student's $t$ test. **f** Subcellular localization of lncCIRPIL in NMCMs and adult mouse cardiomyocytes detected by Fluorescent in situ hybridization (FISH) & Immunofluorescence (IF) after A/R or I/R injury. $n = 3$. Scale bar = 20 μm. DAPI indicates 4′,6-diamidino-2-phenylindole.

from lncCIRPIL-TG mice relative to WT mice (Fig. 3g, h). The Bcl2/Bax ratio was higher in I/R hearts of lncCIRPIL-TG mice than WT mice (Fig. 3i). These data indicated that lncCIRPIL protects the heart from I/R injury by inhibiting apoptosis.

**LncCIRPIL deficiency exacerbates cardiac I/R injury.** We then employed loss-of-function strategy to explore the function of CIRPIL on cardiac I/R injury. The lncCIRPIL global knockout mice (lncCIRPIL-KO mice) were generated by using CRSPR-CAS9 (Clustered Regularly Interspaced Short Palindromic Repeats/CRISPR associated 9) technique (Supplementary Fig. 3). The successful ablation of lncCIRPIL was verified by qRT-PCR (Fig. 4a). Echocardiography showed no significant difference in

the function of hearts between WT and lncCIRPIL-KO mice at the age of 8 weeks. When subjected to I/R injury, the cardiac function of WT mice was impaired as indicated by decreased EF and FS, which was further decreased in lncCIRPIL-KO mice, indicating the aggravation of cardiac function (Fig. 4b). The IA/AAR ratio in lncCIRPIL-KO mice was significantly higher than that of WT mice (Fig. 4c, d). The serum levels of CK-MB and LDH after I/R injury were also significantly higher in lncCIRPIL knockout mice as compared to WT mice (Fig. 4e, f). The level of caspase-3 activity was further increased, and Bcl2/Bax ratio was further decreased in lncCIRPIL-KO mice than WT mice after subjected to I/R injury (Fig. 4g, h). These data indicated that deletion of lncCIRPIL exacerbates cardiac I/R injury by enhancing apoptosis.

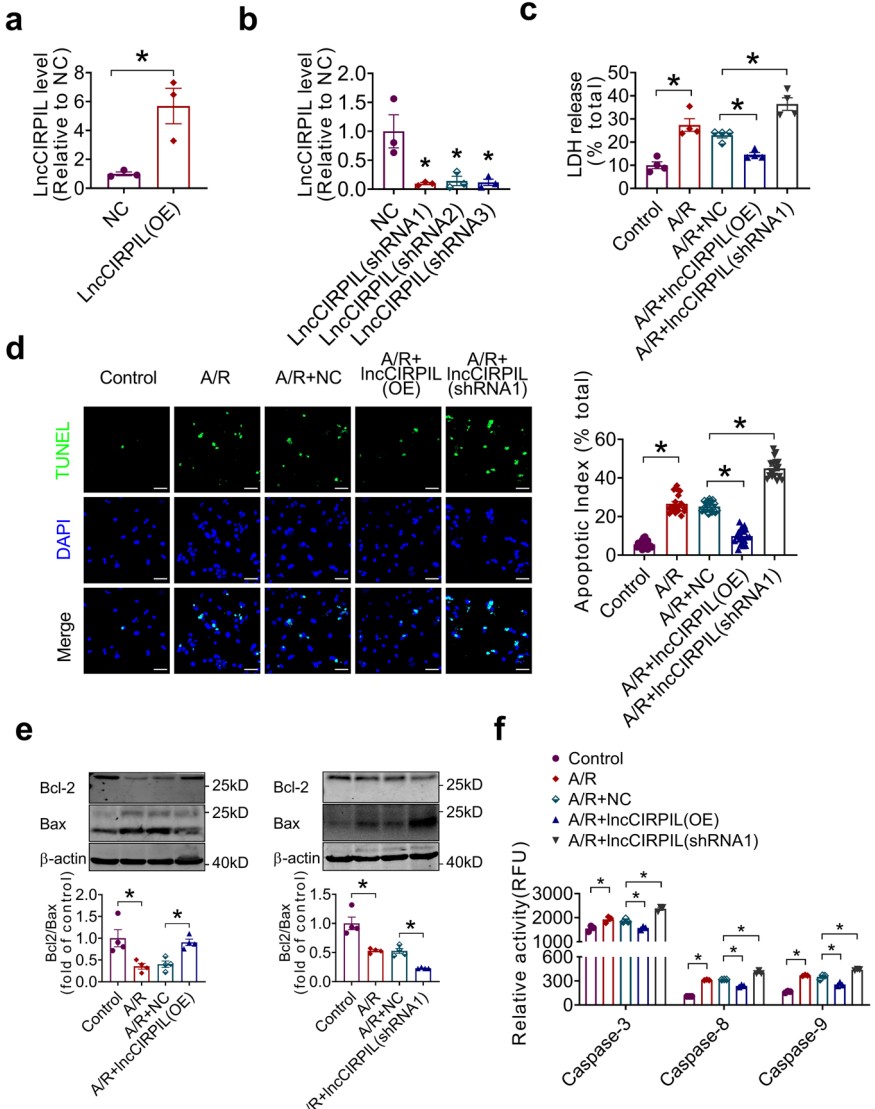

**Fig. 2 Effects of lncCIRPIL on anoxia/reoxygenation(A/R) induced injury of NMCMs. a** Validation of lncCIRPIL overexpression plasmid in NMCMs. $n = 3$. $*P < 0.05$ vs NC (negative control) by two-tailed Student's $t$ test. **b** Knockdown efficiency of shRNAs for lncCIRPIL in NMCMs. $n = 3$. $*P < 0.05$ vs NC by two-tailed Student's $t$ test. **c** Effects of lncCIRPIL on LDH release of NMCMs subjected to A/R insult. $n = 4$. $*P < 0.05$ by one-way ANOVA followed by Tukey post hoc analysis. **d** Effects of lncCIRPIL on cell apoptosis of NMCMs subjected to A/R insult by TUNEL staining. $n = 18$ slices from 3 independent experiments for each group. $*P < 0.05$ by one-way ANOVA followed by Tukey post hoc analysis; Scale bar = 50 μm. **e** Effects of lncCIRPIL on Bcl2/Bax ratio of NMCMs subjected to A/R insult. $n = 4$. $*P < 0.05$ by one-way ANOVA followed by Tukey post hoc analysis. **f** Effects of lncCIRPIL on caspase-3, caspase-8, caspase-9 activities of NMCMs subjected to A/R insult. $n = 3$. $*P < 0.05$ by one-way ANOVA followed by Tukey post hoc analysis.

**LncCIRPIL interacts with p53**. To uncover the molecular mechanisms by which lncCIRPIL mitigates cardiac I/R injury, we performed RNA pull-down with mass spectrometry (MS) assay to identify proteins that interacts with lncCIRPIL. The sense and antisense RNAs of lncCIRPIL were synthesized and biotinylated in vitro, and then incubated with lysates of NMCMs subjected to A/R insult. The precipitated protein complexes were separated on SDS-PAGE gel and performed with silver staining. A sense sequence-specific protein band was collected and subjected to mass spectrometry (MS) analysis and the interacting proteins included p53, PRDX3, NDRG2, HSPA8, HSPA9 (Fig. 5a). As p53 is a classical transcription factor involved in cell apoptosis, we then focused on p53 for validation experiments. Consistently, the bioinformatic analysis by the RNA-Protein Interaction Prediction (RPISeq) database (http://pridb.gdcb.iastate.edu/RPISeq/)[17] also indicated the potential of lncCIRPIL-p53 interaction (Supplementary Table. 6). To validate the lncCIRPIL-p53 interaction, we

reperformed RNA pull-down assay and blotted with anti-p53 antibody. The data showed that the sense sequence, but not the antisense of lncCIRPIL, specifically retrieved p53 from lysates of NMCMs (Fig. 5b) and heart tissues (Supplementary Fig. 5a). Furthermore, RNA immunoprecipitation (RIP) study showed that p53 antibody, but not IgG successfully precipitated lncCIRPIL (Fig. 5c). The same data was obtained when using p53 antibody obtained from a different company, implying the specificity of lncCIRPIL-p53 interaction (Supplementary Fig. 5b, c).

To identify the specific sequence region of lncCIRPIL responsible for binding with p53, three truncated lncCIRPIL fragments (fragment 1,1-220 nts; fragment 2,1-462 nts; and fragment 3, 221-766 nts) were constructed and performed RNA pull-down assay. The data showed that fragment 1 and 2, but not 3 successfully pulled down p53, indicating that the 5′-end fragment (1-220 nts) of lncCIRPIL was the binding region for p53 (Fig. 5d). The functional study showed that fragment 1 and 2

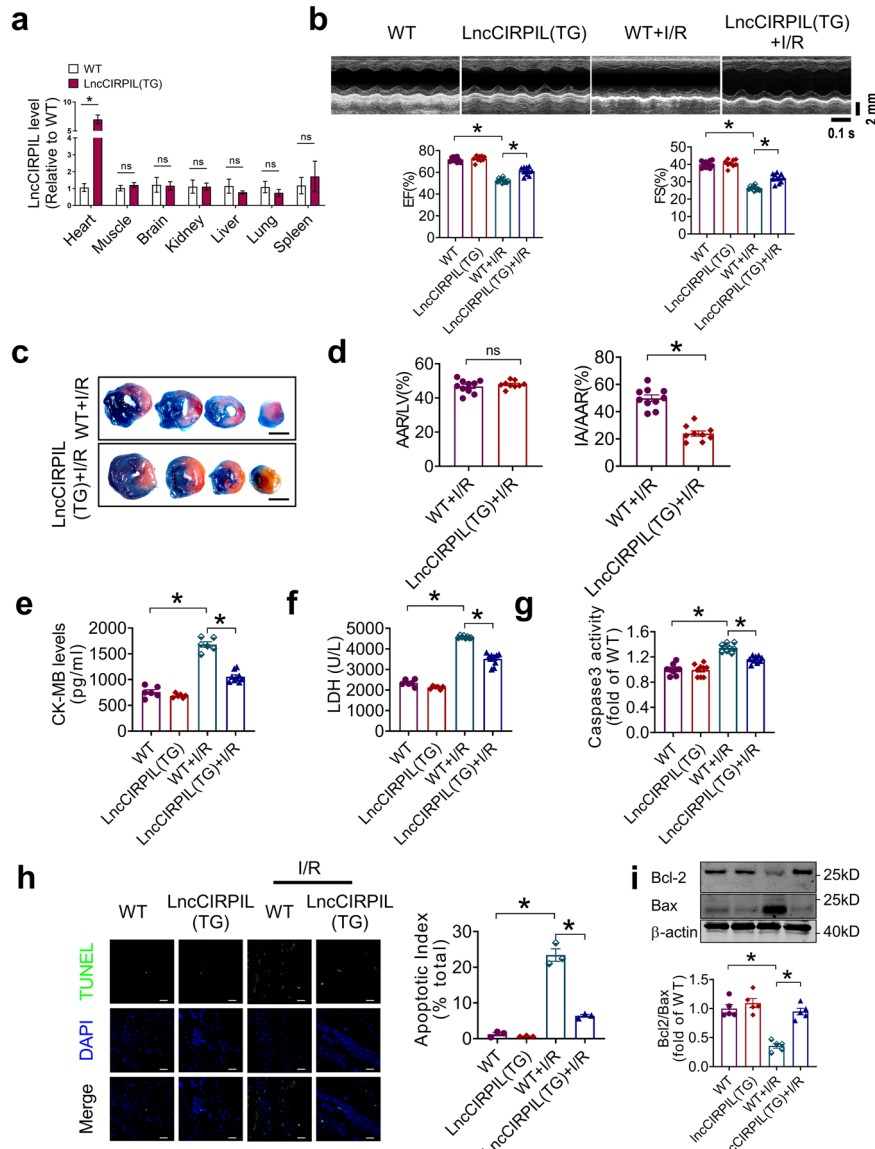

**Fig. 3 Cardiomyocyte-specific overexpression of lncCIRPIL protects the heart from ischemia/reperfusion (I/R) injury in mice. a** LncCIRPIL expression in major organs of WT and lncCIRPIL-TG mice. $n = 3$. *$P < 0.05$ vs WT by two-tailed Student's $t$ test. **b** Representative cardiac echocardiography and statistical analysis of cardiac function of WT and lncCIRPIL transgenic (TG) mice subjected to cardiac I/R injury. EF eject fraction, FS fractional shortening. $n = 9$ to 11. *$P < 0.05$ by one-way ANOVA followed by Tukey post hoc analysis. **c, d** Representative images of heart sections of WT and lncCIRPIL-TG mice after I/R injury by TTC and EVANS blue staining, and statistical analysis of at-risk area (AAR) and infarct area (IA) to AAR ratio. $n = 9$ to 10. *$P < 0.05$ vs WT + I/R by 2-tailed Student $t$ test. scale bar $= 2$ mm. **e** Serum CK-MB in WT and lncCIRPIL-TG mice after cardiac I/R injury. $n = 6$ to 9. *$P < 0.05$ by one-way ANOVA followed by Tukey post hoc analysis. **f** Serum LDH in WT and lncCIRPIL-TG mice after cardiac I/R injury. $n = 6$ to 11 mice. *$P < 0.05$ by one-way ANOVA followed by Tukey post hoc analysis. **g** Caspase3 activity in cardiac tissue of WT and lncCIRPIL-TG mice subjected to I/R injury. $n = 9$. *$P < 0.05$ by one-way ANOVA, followed by Tukey post hoc analysis. **h** Effects of lncCIRPIL on apoptosis of heart tissue from mice subjected to I/R insult by TUNEL staining. $n = 3$. *$P < 0.05$ by one-way ANOVA followed by Tukey post hoc analysis; Scale bar $= 50$ μm. **i** Bcl2/Bax ratio in cardiac tissues of WT and lncCIRPIL-TG mice subjected to I/R injury. $n = 5$. *$P < 0.05$ by one-way ANOVA, followed by Tukey post hoc analysis.

which contain the 1-220 nt sequence of lncCIRPIL, but not 3, inhibited the release of LDH in NMCMs subjected to A/R injury (Fig. 5e).

To further define the binding domains of p53 with lncCIRPIL, HEK293T cells were transfected with Flag-p53 and two Flag-p53 truncated fragments containing either N-terminal or C-terminal domain. The result showed that lncCIRPIL binds to the C-terminal domain of p53 (Fig. 5f). Furthermore, RIP study with Flag antibody showed that C-terminal domain of p53, but not N-terminal domain, successfully immunoprecipitated lncCIRPIL (Fig. 5g).

These data indicated that 5′-end fragment lncCIRPIL interacts with C-terminal domain of p53.

**LncCIRPIL promotes the ubiquitin-mediated degradation of p53 during cardiac injury.** Studies have demonstrated that p53 exerts diverse biological actions depending on its specific subcellular locations[10,18,19]. To explore the potential impact of lncCIRPIL's binding on p53, we examined the distribution of p53 in cardiomyocytes during cardiac injury. The immunofluorescence staining data showed that low basal level of p53 distributed mainly in the cytoplasm of normal cultured NMCMs,

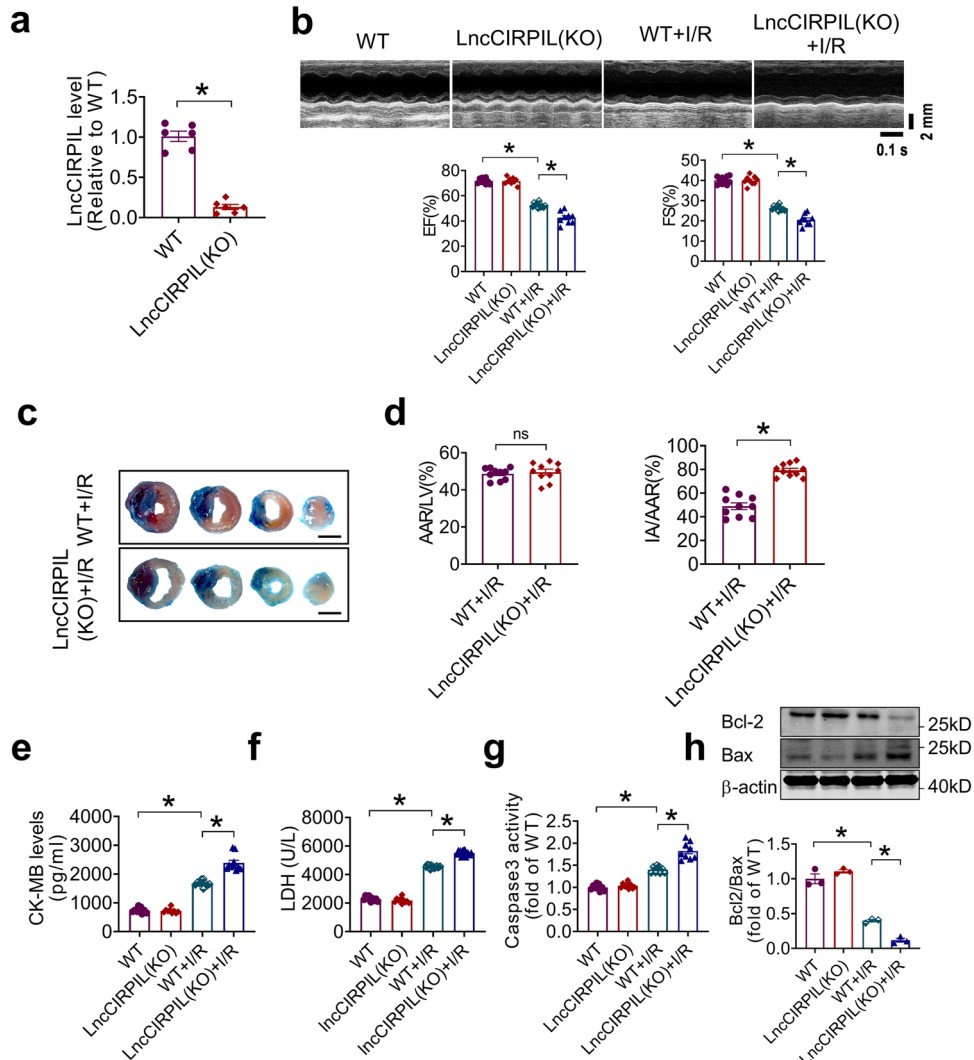

**Fig. 4 Knockout of lncCIRPIL exacerbates cardiac ischemia/reperfusion (I/R) injury in mice. a** The expression of LncCIRPIL in heart tissue of WT and lncCIRPIL-KO mice. $n = 6$. *$P < 0.05$ vs WT by two-tailed Student's $t$ test. **b** Representative cardiac echocardiography and statistical analysis of cardiac function of WT and lncCIRPIL-KO mice subjected to cardiac I/R injury. EF, eject fraction; FS, fractional shortening. $n = 9$ to 11. *$P < 0.05$ by one-way ANOVA followed by Tukey post hoc analysis. **c, d** Representative images of heart sections from WT and lncCIRPIL-KO mice after I/R injury by TTC and EVANS blue staining, and statistical analysis of at-risk area (AAR) and infarct area (IA) to AAR ratio. $n = 10$. *$P < 0.05$ vs WT + I/R by two-tailed Student's $t$ test. Scale bar = 2 mm. **e** Serum CK-MB in WT and lncCIRPIL-KO mice after cardiac I/R injury. $n = 9$ or 10. *$P < 0.05$ by one-way ANOVA followed by Tukey post hoc analysis. **f** Serum LDH in WT and lncCIRPIL-KO mice after cardiac I/R injury. $n = 9$ to 16. *$P < 0.05$ by one-way ANOVA followed by Tukey post hoc analysis. **g** The caspase3 activity in cardiac tissue of WT and lncCIRPIL-KO mice subjected to I/R injury. $n = 9$. *$P < 0.05$ by one-way ANOVA followed by Tukey post hoc analysis. **h** Bcl2/Bax ratio in cardiac tissue of WT and lncCIRPIL-KO mice subjected to I/R injury. $n = 3$. *$P < 0.05$ by one-way ANOVA followed by Tukey post hoc analysis.

which was not affected by overexpression of lncCIRPIL. When subjected to A/R insult, both cytoplasmic and nuclear p53 proteins were dramatically increased, and tended to accumulate in the nucleus, and lncCIRPIL overexpression markedly suppressed this change (Fig. 6a). Consistently, the western blot data showed that the total, cytoplasmic, and nuclear levels of p53 protein were increased in NMCMs with A/R insult, which were inhibited by overexpression of lncCIRPIL (Fig. 6b). The upregulation of *p53* mRNA during A/R insult was not affected by lncCIRPIL overexpression (Fig. 6c). The same qualitative alterations in p53 protein distribution and expression were observed in adult cardiomyocytes isolated from lncCIRPIL-TG mice subjected to I/R injury (Supplementary Fig. 5a–c). In contrast, knockdown of lncCIRPIL in NMCMs further upregulated the cytoplasmic and nuclear expression of p53 (Supplementary Fig. 5d, e).

As lncCIRPIL did not influence the mRNA level of p53 during A/R insult, we speculated that lncCIRPIL may alter p53 protein level at post-translational level. Ubiquitin-mediated degradation is a major mechanism for the turnover of p53 protein[20–22]. We therefore raised the notion that reduction of p53 protein level by lncCIRPIL may be caused by increase of p53 ubiquitination. The data showed that overexpression of lncCIRPIL increased ubiquitination of p53 after A/R injury in NMCMs when the proteasome mediated degradation of proteins is blocked by proteasomal inhibitor MG132 (Fig. 6d). Meanwhile, blocking of proteasome mediated degradation by MG132 led to the accumulation of p53 uniquely in the cytoplasm with little in the nucleus of lncCIRPIL overexpressing NMCMs under A/R insult, indicating that lncCIR-PIL sequesters p53 in the cytoplasm (Fig. 6e). Several E3 ligases such as CHIP, COP1, MDM2, HUWE1 and RCHY1 have been

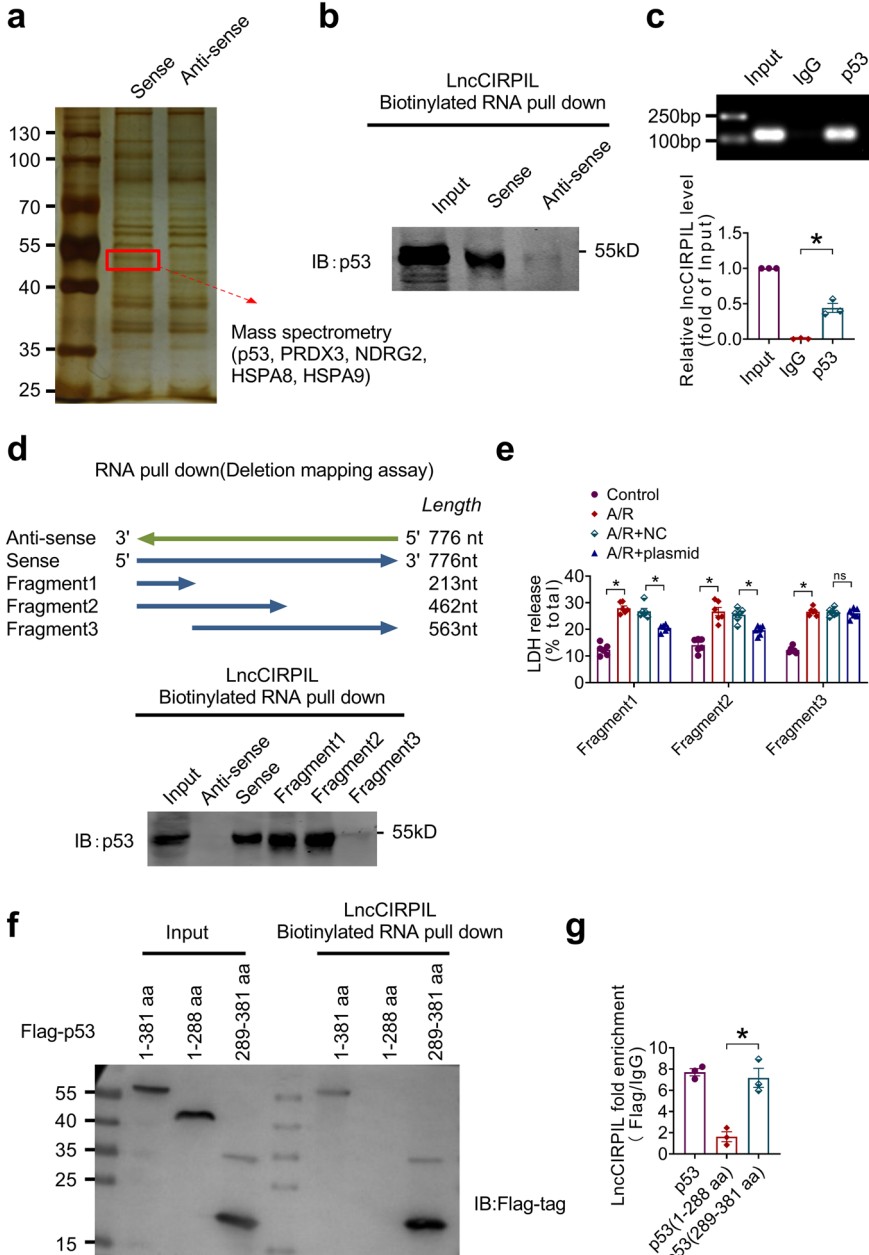

**Fig. 5 LncCIRPIL binds with p53. a** Silver staining of electrophoretic separated proteins pulled down by sense and antisense sequences of lncCIRPIL. Mass spectrometry was performed on the band specifically pulled down by sense sequence and p53 was identified. **b** Western blot of p53 pulled down by sense sequence of lncCIRPIL. $n = 3$. **c** LncCIRPIL immunoprecipitated by anti-p53 antibody. $n = 3$. *$P < 0.05$ vs IgG by two-tailed Student's $t$ test. **d** Western blot of p53 pulled down by biotin-labeled fragments of lncCIRPIL. $n = 3$. **e** Effects of fragments of lncCIRPIL on LDH release of NMCMs subjected to anoxia/reoxygenation(A/R) insult. $n = 6$ samples from 3 independent experiments for each group. *$P < 0.05$ by one-way ANOVA followed by Tukey post hoc analysis. **f** Western blot of flag-labeled fragments of p53 pulled down by sense sequence of lncCIRPIL. $n = 3$. **g** LncCIRPIL immunoprecipitated by anti-flag antibody for flag-labeled fragments of p53. $n = 3$. *$P < 0.05$ by one-way ANOVA followed by Tukey post hoc analysis.

shown to regulate the degradation of p53. We thereby knocked down the expression of these ligases and p53 by their siRNAs (Supplementary Fig. 6a–e). The data showed that knockdown of CHIP, COP1 and MDM2 partially inhibited the degradation of p53 in the cytoplasm of NMCMs under A/R insult (Fig. 6f), indicating the involvement these E3 ligases on the degradation of p53 in this specific condition. Consistent with the distribution change of nuclear p53, lncCIRPIL inhibited the mRNA expression of p53 downstream target genes *Bax*, *Puma* and *Noxa* during A/R insult in NMCMs (Fig. 6g). These data indicated that the binding of lncCIRPIL to p53 promoted the degradation of p53 ubiquitination.

**p53 mediates the regulation of lncCIRPIL on NMCMs injury induced by A/R.** To test whether the protective effect of lncCIRPIL is mediated by p53, the p53 overexpressing plasmid was co-transfected to NMCMs with lncCIRPIL and then subjected to A/R insult. The successful overexpression of p53 was shown in Fig. 7a. Co-transfection of p53 partially abolished the protective effect of lncCIRPIL overexpression in A/R treated NMCMs, as manifested by the increase of LDH release compared to lncCIRPIL group (Fig. 7b). Furthermore, co-transfection of p53 led to an increase in caspase-3 activity and decrease in Bcl2/Bax ratio (Fig. 7c, d). Additionally, the pro-apoptotic effects of knockdown of lncCIRPIL were counteracted

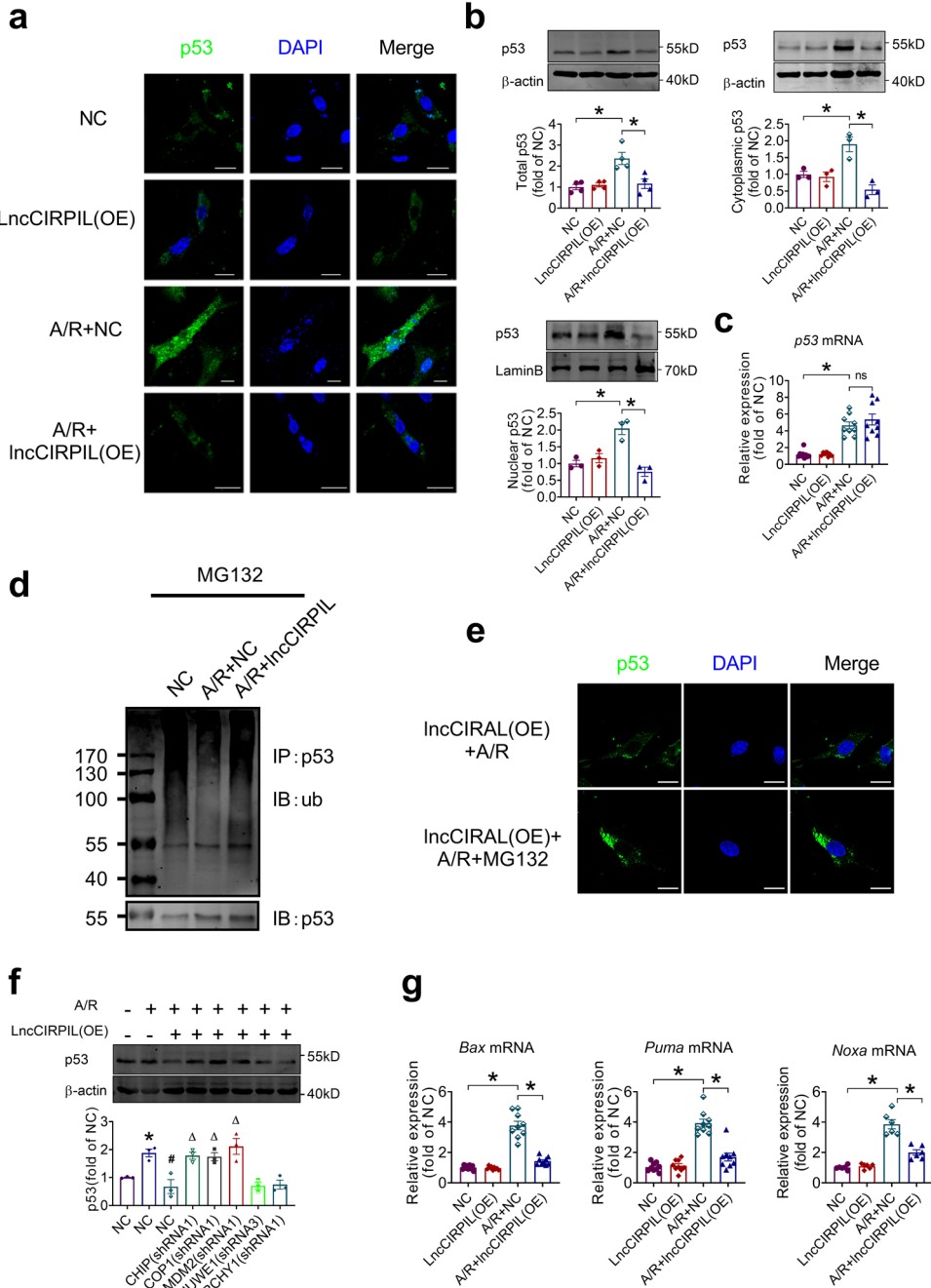

**Fig. 6 LncCIRPIL promotes the ubiquitin-mediated degradation of p53 during cardiac injury. a** Influence of lncCIRPIL overexpression on cytoplasmic and nuclear distribution of p53 in NMCMs subjected to A/R insult by immunofluorescence staining. $n = 3$. Scale bar = 20 μm. **b** Effects of lncCIRPIL overexpression on total, cytoplasmic and nuclear levels of p53 in NMCMs subjected to anoxia/reoxygenation (A/R) insult. $n = 3$. *$P < 0.05$ by one-way ANOVA followed by Tukey post hoc analysis. **c** Effect of lncCIRPIL on the mRNA level of *p53* in NMCMs subjected to A/R insult. $n = 9$ samples from 3 independent experiments. *$P < 0.05$ by one-way ANOVA, followed by Tukey post hoc analysis. **d** Effects of lncCIRPIL overexpression on the ubiquitination of p53 in NMCMs subjected to A/R insult. $n = 4$. **e**. Blocking of protein degradation by MG132 leads to cytoplasmic accumulation of p53 in lncCIRPIL overexpressing NMCMs subjected to A/R insult. $n = 4$. Scale bar = 20 μm. **f** The shRNAs of CHIP, MDM2, COP1, HUWE1 and RCHY1 on the protein level of p53 in NMCMs subjected to A/R insult. $n = 3$. *$P < 0.05$ vs NC, #$P < 0.05$ vs A/R + NC, Δ$P < 0.05$ vs A/R + lncCIRPBL(OE) by one-way ANOVA followed by Tukey post hoc analysis. **g** Effects of lncCIRPIL overexpression on the transcription of p53-downstream genes *Bax Puma* and *Noxa*. $n = 9$ samples for *Bax* and *Puma*, $n = 6$ samples for *Noxa* from 3 independent experiments. *$P < 0.05$ by one-way ANOVA followed by Tukey post hoc analysis.

by p53 siRNA, as manifested by the reduced LDH release, caspase-3 activity, and increased Bcl2/Bax ratio in NMCMs under A/R stimulation (Fig. 7e–g). p53 knockdown efficiency was shown in Supplementary Fig. 6f. These data implied that the protective effect of lncCIRPIL on cardiomyocyte is mediated by p53.

**Effects of human conserved fragment of lncCIRPIL (hcf-lncCIRPIL) on AC16 cells exposed to A/R insult.** We evaluated if lncCIRPIL is conserved across species, especially mice and humans. By using UCSC (University of California, Santa Cruz) Browser, we identified that the genomic locus of lncCIRPIL was

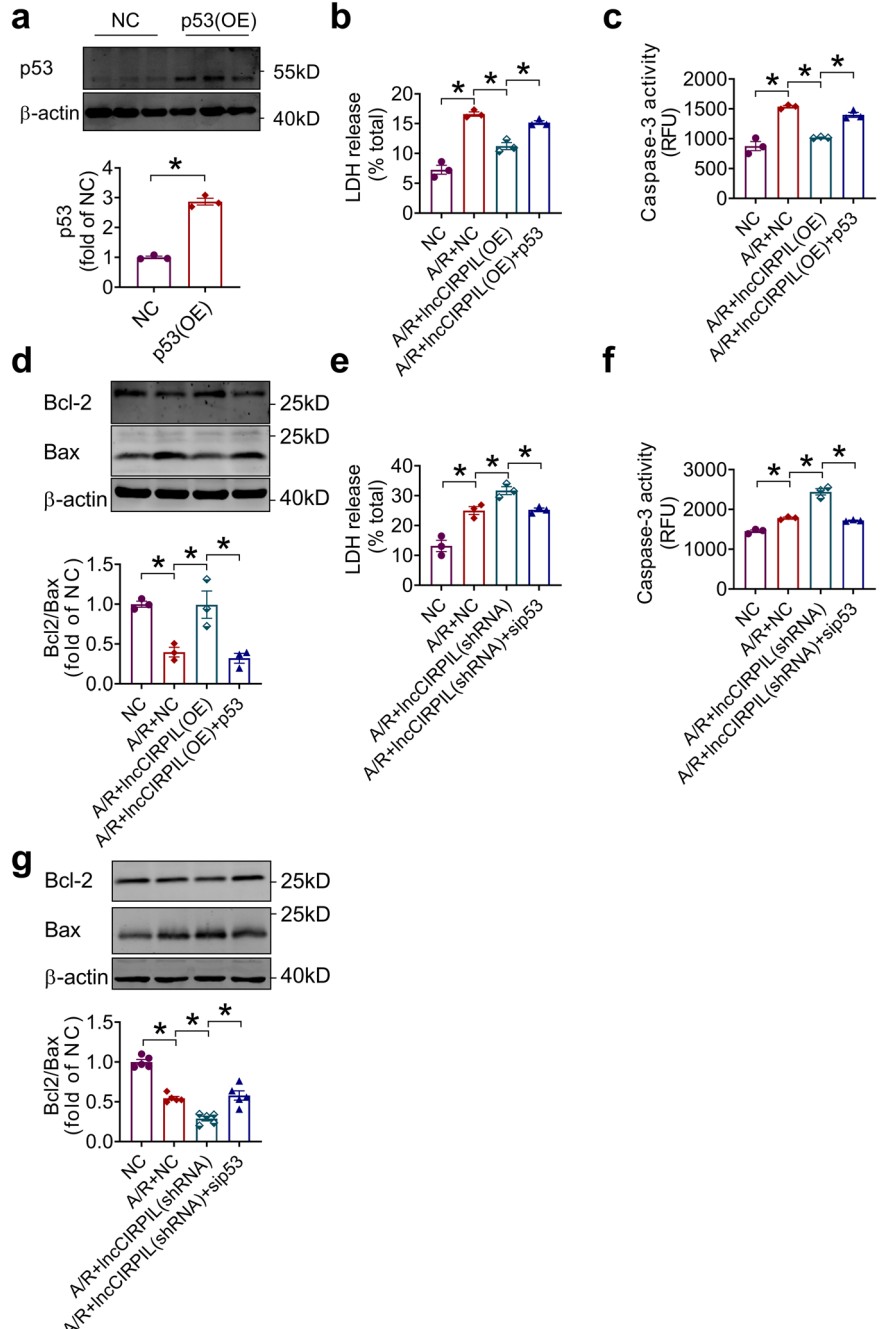

**Fig. 7 p53 mediates the regulation of lncCIRPIL on NMCMs injury induced by A/R. a** Validation of p53 overexpression plasmids in NMCMs. $n = 3$.
*$P < 0.05$ vs NC (negative control) by two-tailed Student's $t$ test. **b** Effect of p53 co-transfection on LDH release of lncCIRPIL overexpressing NMCMs
subjected to A/R insult. $n = 3$. *$P < 0.05$ by one-way ANOVA followed by Tukey post hoc analysis. **c** Co-transfection of LncCIRPIL and p53 plasmids on
caspase-3 activity in NMCMs subjected to A/R insult. $n = 3$. *$P < 0.05$ by one-way ANOVA followed by Tukey post hoc analysis. **d** Co-transfection of
LncCIRPIL and p53 plasmids on Bcl2/Bax ratio in NMCMs subjected to A/R insult. $n = 3$. *$P < 0.05$ by one-way ANOVA followed by Tukey post hoc
analysis. **e** LDH release of NMCMs co-transfected with lncCIRPIL siRNA and p53 siRNA. $n = 3$. *$P < 0.05$ by one-way ANOVA followed by Tukey post hoc
analysis. **f** Caspase-3 activity in NMCMs co-transfected with lncCIRPIL siRNA and p53 siRNA. $n = 3$. *$P < 0.05$ by one-way ANOVA followed by Tukey post
hoc analysis. **g** Bcl2/Bax ratio in NMCMs subjected to A/R insult co-transfected with lncCIRPIL siRNA and p53 siRNA. $n = 3$. *$P < 0.05$ by one-way
ANOVA followed by Tukey post hoc analysis.

highly conserved across species (Supplementary Fig. 7a)[23]. By using
BLAST (Basic Local Alignment Search Tool)[24,25]_ENREF_24, we
found a high degree conservation sequence of lncCIRPIL (17-71
nts) in human genome, for convenience it named as the human
conserved fragment of lncCIRPIL (hcf-lncCIRPIL) (Supplementary
Fig. 7b). We therefore evaluated the function of hcf-lncCIRPIL on
human AC16 cells exposed to A/R insult. The plasmid of hcf-

lncCIRPIL was constructed and first used to detect the interaction
between hcf-lncCIRPIL and p53 in human AC16 cells, and a strong
combine between hcf-lncCIRPIL and p53 was recognized by RNA
pull-down assay (Fig. 8a). Afterwards, the plasmid of hcf-
lncCIRPIL was transfected to human AC16 cells (Fig. 8b). Over-
expression of hcf-lncCIRPIL prevented the decrease of Bcl2/Bax
ratio in human AC16 cells exposed to A/R insult (Fig. 8c).

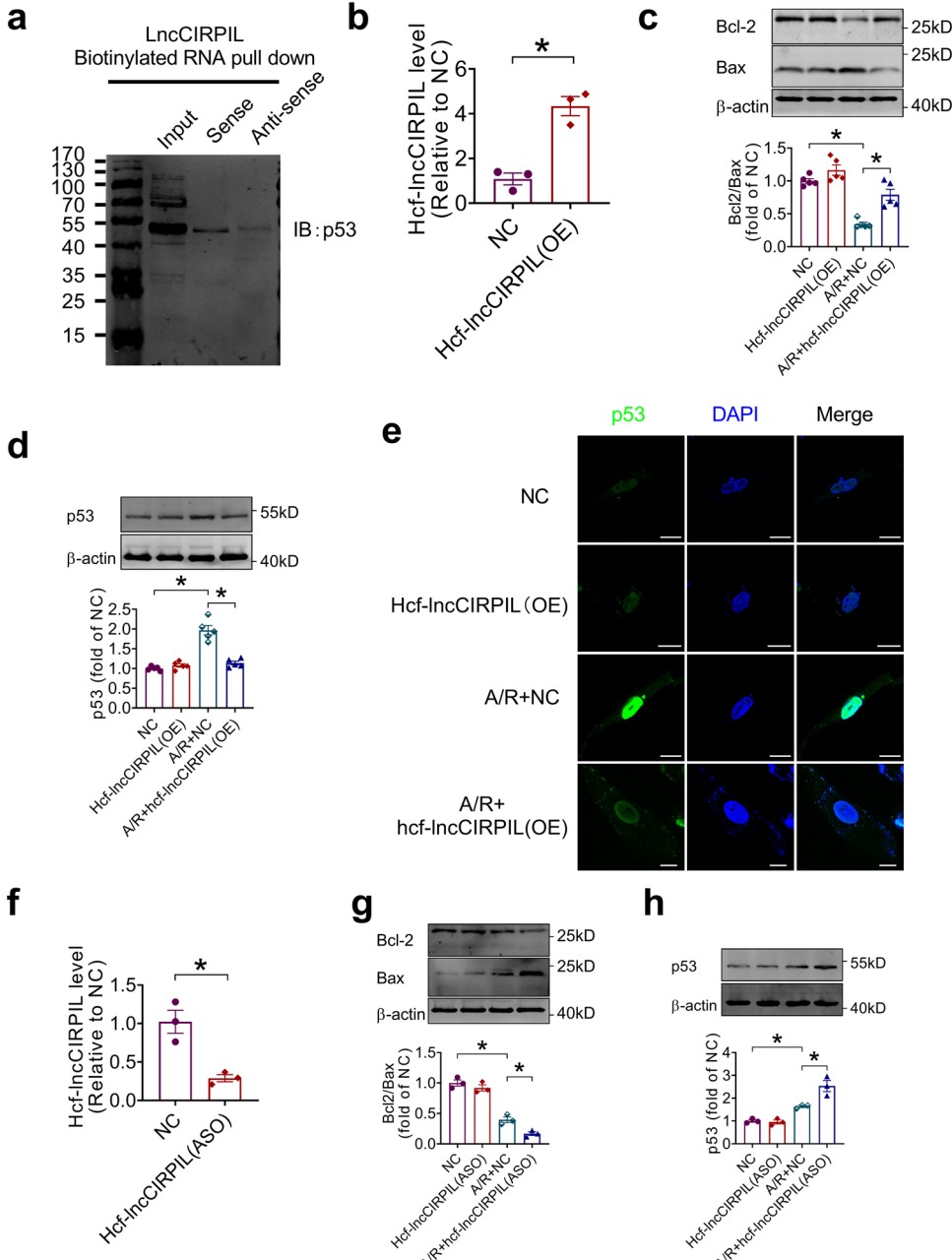

**Fig. 8 The effects of human conserved fragment of lncCIRPIL (hcf-lncCIRPIL) on human AC16 cells exposed to A/R insult. a** Western blot of p53 pulled down by sense sequence of hcf-lncCIRPIL in human AC16 cells. $n = 3$. **b** The level of hcf-lncCIRPIL in AC16 cells transfected with hcf-lncCIRPIL overexpression plasmid. $n = 3$. *$P < 0.05$ vs NC by two-tailed Student's $t$ test. **c**, **d** Effects of hcf-lncCIRPIL overexpression on Bcl2/Bax ratio and p53 level in AC16 cells exposed to A/R insult. $n = 5$. *$P < 0.05$ by one-way ANOVA followed by Tukey post hoc analysis. **e** Effects of hcf-lncCIRPIL overexpression on p53 cellular distribution. $n = 3$. Scale bar = 20 μm. **f**. Knockdown efficiency of hcf-lncCIRPIL by the antisense oligonucleotide (ASO). $n = 3$. *$P < 0.05$ vs NC by two-tailed Student's $t$ test. **g**, **h** Effects of hcf-lncCIRPIL-ASO on Bcl2/Bax ratio and p53 level in AC16 cells exposed to A/R insult by. $n = 3$. *$P < 0.05$ by one-way ANOVA followed by Tukey post hoc analysis.

Upregulation of p53 induced by A/R insult was inhibited by hcf-lncCIRPIL overexpression (Fig. 8d). p53 mainly distributed in the nucleus of AC16 cells at low basal level. It was dramatically increased and accumulated in nucleus of AC16 cells after exposing to A/R insult. Hcf-lncCIRPIL overexpression significantly suppressed the upregulation of p53 induced by A/R insult (Fig. 8e).

We then performed loss-of-function study by knocking down hcf-lncCIRPIL with its antisense oligonucleotides (ASO) (hcf-lncCIRPIL-ASO) (Fig. 8f). Transfection of hcf-lncCIRPIL-ASO further suppressed the Bcl2/Bax ratio, and upregulated p53

expression in human AC16 cells exposed to A/R insult (Fig. 8g, h). These data suggested that the human homologue of lncCIRPIL may exert protecting effect against cardiac I/R injury.

## Discussion

In the present study, we identified a cytoplasmic lncRNA-CIRPIL that acts as a critical regulator in cardiac I/R injury. LncCIRPIL was downregulated during cardiac injury. Overexpression of lncCIRPIL attenuates cardiomyocyte injury by inhibiting

**Fig. 9 Schematic diagram depicting the proposed signaling mechanisms underlying the effects of lncCIRPIL in the setting of I/R.** Upon ischemia/reperfusion stress, the level of lncCIRPIL is decreased and p53 is increased in cardiomyocytes. The reduction of lncCIRPIL sequesters less p53 for ubiquitination mediated degradation. Thereby, more p53 translocate to nucleus to trigger the transcription of target genes to initiate cell apoptosis.

apoptosis in vitro and in vivo. Mechanistically, lncCIRPIL directly binds to p53 and inhibits its nuclear translocation, leading to the transcriptional inhibition of p53 target genes, which thereby alleviated cardiomyocyte injury (Fig. 9). These findings not only shed light on the molecular mechanism of cardiac I/R injury, but also imply the therapeutic potential of noncoding RNAs in cardiac diseases.

LncRNA expression profiles were deregulated in various cardiac diseases, and many deregulated lncRNAs are proved to be critical regulators of specific diseases[15,26,27]. In this study we screened the lncRNA profiles in I/R hearts and identified a series of aberrantly expressed lncRNAs, indicating that the I/R stress is able to interfere the transcription of lncRNAs. The distribution of lncRNA manifests tissue or spatial specificity, which hints the tissue-specific function of lncRNAs[13,14]. The cardiac enrichment of lncCIRPIL and its responsiveness to oxidative stress strongly implies its potential role in the heart. The notion was verified by lncCIRPIL gain- and loss-of function studies. Compared with WT mice, the infarct size of lncCIRPIL-TG mice was significantly reduced and cardiac function was improved after cardiac I/R injury, whereas the opposite was observed in lncCIRPIL-KO mice. During cardiac I/R injury, induce apoptosis can cause damage and loss of cardiomyocyte[28]. In this study, we found that lncCIRPIL inhibited cell apoptosis, and improved cardiac function after I/R injury. Despite of the rapid advancement in elucidating the role of lncRNAs in cardiac I/R injury, few studies employed genetic ablation or overexpression technology to explore the function of lncRNAs[29]. The current study thus provided additional data to confirm the myocardial protective effects of lncRNA.

LncRNAs execute the regulatory functions in heart diseases through versatile mechanisms depending on its subcellular distribution, base sequence, function domain, spatial structure, and expression pattern[30,31]. The current study showed that lncCIRPIL specifically located in the cytoplasm, which arises the possibility that lncCIRPIL may exert its action by *trans* mode. This is also supported by the fact that lncCIRPIL is an intergenic lncRNA. Cytoplasmic lncRNA has been demonstrated to regulate nucleic acid splicing, degradation and act as guides, decoys, signals, or scaffolds[32,33]. Proteins are the ultimate executor of gene functions, and they represent the large portion of the interactome of lncRNAs, with a great number of lncRNAs achieving their regulatory role through interacting with proteins[34,35]. We analyzed the proteins binding to lncCIRPIL and discovered that p53

directly interacts with lncCIRPIL. p53 is a master transcription factor that suppresses tumor development, which is deeply involved in the regulation of cell apoptosis[20,36,37]. p53 exacerbates cardiac apoptosis during cardiac I/R injury[11,38]. The protein level and biological activity of p53 are strictly regulated at multiple levels. When exposed to stress such as hypoxia the protein level of p53 is rapidly elevated, but under physiological condition it is maintained low in the heart[39]. Our observation that lncCIRPIL reduced the protein level of p53 and sequestered it in the cytoplasm explains its inhibitory role on cardiomyocyte apoptosis. The notion was further confirmed by that fact that p53 overexpression abrogated the protective effects of lncCIRPIL, and knockdown of p53 counteracted the pro-apoptosis effects of lncCIRPIL(shRNA).

It is known that post-transcriptional regulation by ubiquitination is essential for the degradation of p53 protein when it translocates from the nucleus to cytoplasm[21,40]. In this study, lncCIRPIL promoted the ubiquitination of p53 by enhancing its cytosol distribution. This finding indicates that lncCIRPIL acts as an anchor to restrict p53 in the cytoplasm, which initiates the ubiquitinated degradation of p53. There is only one previous study showing the regulation of lncRNA on p53 in cardiac injury, in which study the interaction of lncRNA CAIF and p53 occurs in the nucleus and the transcriptional activity of p53 was inhibited[39]. In this study, LncCIRPIL is a newly discovered cardiac protective factor by inhibiting apoptosis in myocardial ischemia reperfusion (I/R) injury and there was a strong and veritably binding between LncCIRPIL and p53. On this basis, we found that lncRNA-mediated protein modification of p53, specifically ubiquitin degradation inhibited apoptosis of cardiomyocytes. Despite of the critical role of p53 in cardiac I/R injury, previous study showed that deletion of p53 triggered the spontaneous pathological hypertrophy in aged mice, demonstrating that the intervention through p53 to reduce I/R injury requires better strategies[41]. Our results show that lncCIRPIL directly binds to p53 and reduces its nuclear protein level after I/R injury. This may provide a new safety entry point of interfering lncCIRPIL for the intervention of p53 and I/R injury.

In conclusion, we discovered that lncCIRPIL protects the heart from ischemia/reperfusion injury by sequestering p53 in the cytoplasm and promotes its ubiquitin-mediated degradation. The current study highlights the novel regulatory role of lncRNA on p53 activity and the molecular mechanism for the pathogenesis of cardiac I/R injury.

## Methods

**Animals**. Male C57BL/6J mice (8 to 10 weeks old) were provided by the Animal Center at the Second Affiliated Hospital of Harbin Medical University. The use of animals was approved by the Ethic Committees of College of Pharmacy, Harbin Medical University and complied with the Guide for the Care (HMUIRB3005719) and Use of Laboratory Animals published by the US National Institutes of Health (publication No. 85-23; revised, 1996).

**MTA 1.0 transcriptome microarray assay**. The microarray assays were performed by Gminix Company (Shanghai, China). Briefly, total RNA was extracted from heart tissues by using Rneasy Mini Kit (QIAGEN, Germany, 217004) and 100 ng RNA was used to generate fragmented biotinylated cDNA using GeneChip WT Terminal Labeling Kit and Controls Kit (Affymetrix Inc, USA, 901525) according to the manufacturer's instructions. Affymetrix Mouse Transcriptome Array 1.0 (Affymetrix Inc, USA) was hybridized with labeled sample at 45 °C for 16 h in GeneChip Hybridization Oven 640 (Affymetrix Inc, USA). GeneChips were scanned by GeneChip2 Scanner 3000 7G (Affymetrix Inc, USA) and quantified with image and signal intensities. The data was analyzed by the RMA algorithm in Affymetrix Expression Console Software v1.0 for normalization. The microarray data had been uploaded to GEO (Gene Expression Omnibus) public database under the accession number GSE161151.

**Generation of lncCIRPIL cardiac myocyte-specific overexpressing mice**. To generate cardiomyocyte-specific lncCIRPIL overexpressing mice, the expression

vector with the α-MHC (α-Myosin Heavy Chain) promoter and the DNA sequence of lncCIRPIL were constructed and injected into the C57BL/6 mice zygote by prokaryotic microinjection. Then, the DNA sequence of lncCIRPIL was integrated into the mouse genome to achieve the cardiomyocyte-specific lncCIRPIL over-expressing mice (Cyagen, Co, Ltd, China). Genomic DNA was extracted from tail tissue for PCR amplification to identify the lncRNA-CIRPIL transgenic offspring. The primers used for verifying transgenic gene in mice are: forward 5′-AGAGC-CATAGGCTACGGTGTA-3′ and reverse 5′-AGGAGGAACTGAAGATGTCA-GAA-3′.

**Generation of lncCIRPIL knockout mice**. LncCIRPIL conventional knockout mice were generated by using the CRISPR/Cas9 system. The guide RNA (gRNA) and Cas9 plasmids of lncCIRPIL were designed and constructed to knock out the whole sequence of lncCIRPIL, and to obtain the lncCIRPIL conventional knockout mice model in a C57BL/6 background (Cyagen, Co, Ltd, China). The primers used to identify the offspring are: forward 5′-GGAAGACTCAGGGTGCGTACTCAGT-3′ and reverse 5′-CATATACACGCGCACACAGGAGAAT-3′.

**Establishment of cardiac ischemia/reperfusion injury in mice**. Mice (8-10 weeks age, 25 ± 3 g weight) were anesthetized by intraperitoneal injection with 2% avertin (0.2 g kg$^{-1}$, Sigma-Aldrich, St Louis, USA) and artificially ventilated by using a small animal ventilator (UGO BASILE, Biological Research Apparatus, Italy) at the respiratory rate of 100 breaths min$^{-1}$ with a tidal volume of 0.3 ml. The chest was opened by making an incision in the third intercostal space and the heart was exposed. The left anterior descending coronary artery (LAD) was ligated using a 7-0 silk thread, together with a PE tube of 2–3 mm in diameter placed in the knot under a dissecting microscope. If the ST-segment elevation appeared in electrocardiogram, the model was successfully created. After 45 min, the PE tube was removed, and reperfusion was confirmed by observing that the left ventricular anterior wall returned to pink after 15–20 s. The animals were allowed for reperfusion for 24 h. The sham operated group received a suture at the same position without ligation.

**Determination of infarct size**. After 24 hours' reperfusion, the heart was exposed and the coronary artery was re-ligated at the same location. Evans blue dye (2%, Solarbio, China, E8010-5) was intravenously injected to distinguish between ischemic and non-ischemic areas. The heart was then quickly removed and washed in PBS solution. The heart was cut into 1 mm slices and incubated with 2,3,5-triphenyltetrazolium chloride (TTC) (2%, Solarbio, China, G3005) at 37 °C for 40 min. The infarct tissue was white, and the live tissue was stained red. The pictures of slices were taken by the Zeiss microscope (SteREO Discovery. V8, Germany). The area at risk, non-ischemic and infarct areas were analyzed by using Image pro-plus 6.0 software.

**Echocardiography**. Mice under anesthesia were fixed in supine position on 37 °C constant temperature bench. The probe is perpendicular to the left edge of the sternum and is at an angle of 10~30 degrees to the long axis of the body. Under two-dimensional image-guided M-mode echocardiography, more than 10 consecutive cardiac cycles were recorded using Vevo2100 Imaging System (VisualSonics, Toronto, Canada). The left ventricular (LV) parameters including left ventricular end diastolic volume (LVEDV), left ventricular end systolic volume (LVESV), left ventricular internal dimension at end diastole (LVIDd), and left ventricular internal dimension at systole (LVIDs) were measured based on M-mode recordings. The data are presented as the average of measurements of three beats and showed in Supplementary Table 2. And the indices of LV ejection fraction (EF) and LV fractional shortening (FS) were recorded.

**Isolation of adult mouse cardiomyocytes**. The isolation of adult mouse cardiomyocytes was performed as described previously[42]. Mice were anesthetized by intraperitoneal injection of 2% avertin (0.2 g kg$^{-1}$). Hearts were rapidly removed and suspended on langendorff device for retrograde perfusion. After 3–5 min perfusion with calcium-free Tyrode's solution, the heart was perfused with Tyrode's solution containing Type II collagenase (1 mg ml$^{-1}$, Gibco, USA, 17101015), proteinase (0.02 mg ml$^{-1}$, Sigma-Aldrich, USA, P8038) and bovine serum albumin V (BSA, 1 mg ml$^{-1}$, BioFroxx, Germany, 4240GR100) for 15–20 min. When the myocardial tissue become soft, the left ventricle was collected in calcium-free Tyrode's solution and cardiomyocytes were dissociated by gently pipetting the tissue. The isolated single cardiomyocytes were finally equilibrated in Tyrode's solution with 0.18 mM CaCl$_2$ and 1% BSA. Tyrode's solution (in mM): NaCl 137, KCl 5.4, NaH$_2$PO$_4$ 0.16, glucose 10, CaCl$_2$ 1.8, MgCl$_2$ 0.5, HEPES 5.0, and NaHCO$_3$ 3.0 (pH 7.4 adjusted with NaOH).

**Neonatal mice cardiomyocytes culture and treatment**. The neonatal mouse cardiomyocytes were isolated and cultured as previously described[43]. C57BL/6 mice (1~3 days old) were decapitated with scissors and the heart was gently removed to a petri dish on ice with Dulbecco's Modified Eagle Medium (DMEM) (Biological Industries, Israel, 01-052-1A). The atrium was discarded and the ventricle was cut into small pieces. The tissue block was added with trypsin (Beyotime,

China, C0202) and gently shook for a few seconds to dissociate the cells repeatedly. The cells were obtained by centrifuged at 1000 rpm min$^{-1}$ for 5 min and resuspended with culture solution. After culturing in petri dish for 90 min, the medium containing cardiomyocytes was placed in a new culture plate for experiments. To establish anoxia/reoxygenation (A/R) injury model, the culture medium was firstly switched to serum-free medium and incubated in the hypoxia incubator (5% CO$_2$, 95% N$_2$) for 12 h, and then cultured with serum-containing medium under normal condition (95% O$_2$, 5% CO$_2$) for 24 h.

**Culture of AC16 human adult ventricular cardiomyocyte cell line**. The human AC16 cell line was purchased from American type culture collection and cultured as described previously[44]. Briefly, AC16 cells were maintained in Dulbecco's Modified Eagle Medium/Nutrient Mixture F-12 (DMEM/F-12) (Hyclone, USA, SH30023.01) containing 10% fetal bovine serum and 1% penicillin-streptomycin and grown in a water-saturated atmosphere at atmospheric oxygen levels and 5% CO$_2$. When reached 70-80% confluence, cells were deprived of serum and undergo anoxia/reoxygenation injury as demonstrated in NMCMs model.

**HEK293T cell line culture**. HEK293T cell line was purchased from American type culture collection and cultured in DMEM (Biological Industries, 01-052-1A) supplemented with 10% fetal bovine serum (FBS) (Biological Industries, Israel, 04-011-1A/B) and 1% penicillin/streptomycin (Beyotime, China, C0222) at 37 °C with 5% CO2. Cells were transfected with CIRPIL, Flag-p53 or Flag truncations of p53 when reached 70-80% confluence.

**Cell transfection**. Plasmid vectors containing mouse lncCIRPIL cDNA, mouse truncated fragment of lncCIRPIL cDNA, human lncCIRPIL cDNA, mouse p53 cDNA, mouse truncated fragment of p53 cDNA, mouse lncCIRPIL shRNA, mouse CHIP shRNA, mouse COP1 shRNA, mouse MDM2 shRNA, mouse HUWE1 shRNA and mouse RCHY1 shRNA were commercially synthesized by Genechem Co.Ltd (Shanghai, China). Mouse p53 siRNA commercially synthesized by Genechem Co.Ltd (Shanghai, China). The antisense oligonucleotide (ASO) for human lncCIRPIL was designed by Guangzhou RiboBio Co., Ltd. (Guangzhou, China). NMCMs and cell lines were transfected with a final concentration of 2.5 mg L$^{-1}$ of plasmid vectors, shRNA, siRNA or ASO by lipofectamine 2000 reagent (Invitrogen, Carlsbad, America) in opti-DMEM circumstance, respectively. Sequences of shRNAs and siRNAs were shown in Supplementary Table 3.

**RNA-protein pulldown assay**. RNA pulldown assays were performed as described previously[45]. Briefly, lncCIRPIL cDNA plasmid or truncated lncCIRPIL cDNA plasmid were used to synthesize biotinylated transcripts as template. The transcripts were labeled by RNA 3′ End Desthiobiotinylation Kit (Roche, Switzerland, 20163) according to the manufacturer's instructions. NMCMs were undergo A/R insult. HEK293T were transfected with Flag-p53 or Flag truncations of p53 for 24 h. Cells were harvested and lysed by RIPA buffer (Beyotime, China, P0013) supplemented with protease and phosphatase inhibitor cocktails (Beyotime, China, P1011) and anti-RNase reagent (Invitrogen, USA, 10777-019). The magnetic beads conjugated with streptavidin (Invitrogen, USA, 65001) were mixed with the biotinylated RNA and lysate, and the mixture were flip mixed for 1 h at room temperature. Then the magnetic beads were separated and washed 3 times with lysis buffer. The magnetic beads were boiled with SDS loading buffer (Beyotime, China, P0015L) for 10 min. The binding protein was separated by SDS PAGE and the protein bands were silver-stained. Bands of interest were analyzed by mass spectrometry (MS) and further confirmed by western blot.

**RNA isolation and quantitative real-time PCR (qRT-PCR)**. The TRIzol reagent (Invitrogen, USA, 15596026) was used for RNA extraction from NMCMs or myocardial tissues. DNase-treated RNA was reverse transcribed using Trans-Script All-in-one First-strand cDNA Synthesis Supermix for qPCR Kit (TransGen Biotech, China, AT301-02). qRT-PCR was performed using SYBR Green Master (Roche, Switzerland, 04913914001). The relative expression levels were calculated based on Ct values and were normalized to the β-actin levels of each sample. The primers used were listed in Supplementary Table 3.

**RNA-interacting protein immunoprecipitation (RIP)**. RNA immunoprecipitation (RIP) experiments were performed with the Magna RIP RNA-Binding Protein Immunoprecipitation Kit (Millipore, USA, 17-700) following the manufacturer's instructions. Briefly, NMCMs were undergo A/R insult. HEK293T were transfected with lncCIRPIL, Flag-p53 or Flag truncations of p53 for 24 h. Cells were harvested and lysed by RIP lysis buffer. Cell lysates were incubated with p53 antibody or Flag antibody (Supplementary Table 5) at 4 °C overnight. Next, Dynabeads conjugated with streptavidin was added and incubated with the mixture for another 4 h. Then the Dynabeads were magnetic separated and washed 3 times with RIP lysis buffer, p53 binding RNAs were eluted and quantified. The qRT-PCR was used to examine coimmunoprecipitated lncCIRPIL.

**Western blot**. Western blot was performed as previously described[43]. Briefly, 72 h after transfection, the neonatal mouse cardiomyocytes (NMCMs) were harvested.

The total protein was extracted from cells or tissues with RIPA lysis buffer (Beyotime, China, P0013B) and a protease inhibitor cocktail (Beyotime, China, P1011) at 4 °C. NE-PER nuclear and cytoplasmic extraction reagents (Thermo Fisher Scientific, USA, 78835) was used for nuclear and cytoplasmic protein exaction. Then Cell lysate were fractionated by SDS-PAGE (10~15% poly-acrylamide gels) and transferred to Pure Nitrocellulose Blotting Membrane (Pall Corporation, USA, S80209). The membranes were blocked in Tris-buffered saline (TBS) containing 5% BSA and then incubated with primary antibodies at 4 °C overnight. The β-actin and LaminB were used as internal controls. After washed with TBS with 0.5% Tween 20 (TBST), the membranes were incubated with the secondary antibody for 60 min at room temperature. The membranes were scanned and analyzed by ODYSSEY machine (LI-COR, USA). The antibodies used were listed in Supplementary Table 5.

**Flow cytometry**. To quantify cell death, NMCMs were harvested by digesting with 0.05% trypsin without EDTA after washing in cold PBS. Proteolysis was then neutralized with fetal bovine serum, and the cell suspension were concentrated and resuspended in 100 μL PBS. Annexin V-FITC and PI staining solution (Beyotime) were added, and the suspension was subjected to flow cytometry (Backman Coulter, USA, CytoFLEX). Live NMCMs were set on the FSC/SSC dot pattern of unstained control cells (Supplementary Fig. 8a). And the singlet population were further confirmed by FSC-H/FSC-A dot pattern of unstained control cells (Supplementary Fig. 8b). Three cell subpopulations identified from the singlet population of Annexin V-FITC & PI stained A/R cells with unstained control cells as reference (Supplementary Fig. 8c, d).

**RNA fluorescence in situ hybridization (FISH)**. The Fluorescent In Situ Hybridization Kit (RiboBio, China, C10910) was used for in situ hybridization of lncCIRPIL. Briefly, a series of pretreatments including fixing, washing, permeating, and drying were performed on the cardiomyocyte and then the probe designed for the target RNA were used to perform hybridization. The secondary antibody was then used to amplify the probes that hybridized to the lncCIRPIL. The total nuclei were stained by DAPI. After mounted with a mounting solution, the staining was observed under a confocal microscope (Zeiss LSM810, Germany).

**Immunofluorescence**. Isolated cardiomyocytes were attached to the adhesive slide. The cells were rinsed with cold PBS for three times and fixed in 4% paraformaldehyde (Beyotime, P0099) for 15 min. After rinsed with PBS for three times, the cells were incubated with 0.5% Triton X-100 (Beyotime, China, P0096) for 1 h at room temperature, and blocked with 10% normal goat serum (Merck, USA, G9023) for 1 h at 37 °C. Then, p53 antibody (1:500) was added into each slide and incubated at 4 °C overnight. After rinsed with PBS for three times, the fluorescein-488 antibody (Merck, USA, SAB4600234) (1:500) were added and the samples were incubated at room temperature for 1 h in a wet box. After rinsing with PBS for three times, the nuclei were counterstained with DAPI for 15 min at room temperature. Immunofluorescence was analyzed under a confocal microscope (Zeiss LSM810, Germany).

**Lactate dehydrogenase (LDH) detection**. The LDH levels in the incubation medium (released LDH), supernatant of cell lysate (retained LDH) of NMCM and serum were determined by spectrophotometry according to the manufacturer's instructions (Nanjing Jiancheng Bioengineering Institute, China, #A020-2-1). The NMCM released level are shown as the percentage of released LDH compared to the total (released plus retained) LDH as described previously[46].

**Creatine kinase MB detection**. The level of creatine kinase MB in mice blood were examined by using CK-MB (Creatine Kinase MB Isoenzyme) ELISA Kit (Elabscience, China, E-EL-M0355c) according to the manufacturer's instructions. Briefly, add 100 μL standard solution or serum sample into the corresponding Micro ELISA plate and incubate at 37 °C for 90 min. Discard the liquid and add 100 μL biotinylated detection antibody working solution immediately, incubate at 37 °C for 60 min. After discarding the liquid and washing the Micro ELISA plate, add 100 μL HRP conjugate working solution and incubate for 30 min at 37 °C. Discard the liquid and wash Micro ELISA plate, add 90 μL substrate reagent and incubate at 37 °C for 15 min. Finally, add 50 μL stop solution and read at 450 nm.

**TUNEL staining**. TUNEL (Terminal deoxynucleotidyl transferase dUTP nick end labeling) staining was performed with the CardioTACSTM in situ apoptosis detection kit (Roche Applied Science, Switzerland, 11684795910) according to the manufacturer's instructions. The total nuclei were counterstained with DAPI for 15 min at room temperature. Immunofluorescence was analyzed under a confocal microscope (Zeiss LSM810, Germany).

**Caspase-3, caspase-8, caspase-9 activity detection**. For NMCM cells, the CASP3 Activity Assay Kit (Cell signal tec, USA, #5732) was used according to the manufacturer's protocol. Briefly, the assay was based on detection of cleavage of fluorogenic substrate Ac-DEVD-AMC (N-Acetyl-Asp-Glu-Val-Asp-7-amino-4-methylcoumarin) for caspase-3. During the assay, the highly fluorescent products

AMC cleaved by caspase-3 can be detected using the fluorescence reader (excitation/emission = 380/460 nm). For heart tissue samples, the CASP3 Activity Assay Kit (Abcam, USA, #ab39383) was used according to the manufacturer's protocol. Briefly, tissue lysate (50 μg) was added to the fluorogenic substrate DEVD-AFC (AFC: 7-amino-4-trifluoromethyl coumarin) for caspase-3, and incubated at 37 °C for 2 h. DEVD-AFC emits blue light (λ max = 400 nm); free AFC emits a yellow-green fluorescence (excitation/emission = 400/505 nm), which can be quantified by spectrophotometry using the fluorescence reader.

To detect the activity of caspase-8, caspase-9, another detection kit was used (Abcam, USA, #ab219915) according to the manufacturer's protocol. Briefly, caspase assay solution was added to cells and incubated at room temperature for 30–60 min.

The final caspase-8 activity was detected by Ex/Em = 535/620 nm; and the final caspase-9 activity was detected by Ex/Em = 370/450 nm.

**Statistics and reproducibility**. The experimental data are expressed as mean ± SEM. One-way analysis of variance (ANOVA) followed by Tukey's post hoc multi-comparison test was used for comparison between multiple groups. Two-tailed Student's $t$ test was used for comparison between two groups. $P < 0.05$ indicates that the difference is statistically significant.

**Reporting summary**. Further information on research design is available in the Nature Research Reporting Summary linked to this article.

## Data availability

Source data can be found in Supplementary Data 1. Other data supporting the findings of this study are available from the corresponding author upon request. The lncCIRPIL microarray data are deposited to the Gene Expression Omnibus (GSE) with the accession number GSE161151. Uncropped and unedited blots are provided in Supplementary Information (Unprocessed images used for western blot assay; unprocessed images used for RNA-interacting protein immunoprecipitation assay).

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

## Acknowledgements
This work was supported by the National Key R&D Program of China [2017YFC1307404 to Zhenwei Pan], National Natural Science Foundation of China [81730012, 81861128022 to B.Y., 82070344, 81870295 to Z.P., 82104433 to Y.J.], Heilongjiang Touyan Innovation Team Program and CAMS Innovation Fund for Medical Sciences [CIFMS, 2019-I2M-5-078 to B.Y.], Guangdong Basic and Applied Basic Research Foundation [2020A1515110050 to Y.J.], Medical Scientific Research Foundation of Guangdong Province of China [A2021006 to Y.J.].

## Author contributions
Y.J., Y.Y., and Y.Z. performed experiments, analyzed data, and prepared the manuscript. J.Y., M.Z., S.L., G.X., X.L., X.Z., J.Y., X.H., and Q.H. helped perform experiments and collect data. B.Y. and Y.L. oversaw the project and proofread the manuscript. H.S. proofread the manuscript. Z.P. designed the project, oversaw the experiments and prepared the manuscript.

## Competing interests
The authors declare no competing interests.
