## [Peer Review File · Communications Biology]

Reviewers' comments:

Reviewer #1 (Remarks to the Author):

This study reported the identification of a new cardiac-enriched lncRNA, lncCIRPIL, and examined its role in protect heart against I/R injury. The underlying mechanisms were also elucidated. The study is well designed, and presented data strongly support the cardiac protection of lncCIRPIL. One concern is that previous studies have shown that inducing autophagy blunts I/R injury (10.1161/CIRCULATIONAHA.113.002416), but this contradictory was not sufficiently discussed. Other specific comments are listed below.

1. How is the expression of lncCIRPIL in I/R patients? This data will strongly increase the clinical relevance of the present study.
2. Introduction, "tumor suppressor p53 functions as a master 63 regulator of cell apoptosis and autophagy." Please add references here.
3. Figure 2, please also measure the activities of caspase-8 and caspase-9. In addition, activation of autophagy should be verified by accurately measuring autophagy flux using an autophagy inhibitor.
4. Figure 3. The expression level of lncCIRPIL in other major organs of the lncCIRPIL-TG mice should be shown to confirm cardiac specificity.
5. Figure 6a. the quality of images were poor to support the statement that p53 level increased in nuclear of A/R+NC cells.
6. Figure 6b showed that lncCIRPIL reduced A/R-induced p53 protein over-expression in both cytosol and nuclear. But the data is not sufficient to support the conclusion that "In this study, lncCIRPIL promoted the ubiquitination of p53 by enhancing its cytosol distribution" line 333.
7. The effect of lncRNA, such as lncRNA-CAILin regulating p53 and autophagy and cardiomyocyte damage during I/R, and the mechanism by which P53 regulating autophagy and cell apoptosis has been established. The authors should further discuss the mechanistic novelty of the present study.
8. Why overexpression of lncCIRPIL increases ubiquitination of p53 after A/R injury but not in controls should be explained.
9. Previous studies showed that inducing cardiomyocyte autophagy blunt I/R injury (10.1161/CIRCULATIONAHA.113.002416), which is contract to the current findings. This discrepancy should be discussed.

Reviewer #2 (Remarks to the Author):

In this manuscript, Jiang, Yang, and Zhang et al. identified a new lncRNA (CIRPIL) through microarray screening with cardiac ischemia/reperfusion (I/R) tissues. The authors went on to prove the role of CIRPIL as a beneficial player in cardiomyocytes under reperfusion injury in vitro. Both loss- and gain-of-function approaches have been employed. Furthermore, the authors generated transgenic and knockout mouse models to validate its role in vivo under cardiac I/R. Next, the authors identified a direct target of this lncRNA as p53. Mechanistically, lncRNA-CIRPIL stimulated the degradation of p53, thereby improving cardiomyocyte survival under I/R. Taken together, the authors carefully established a lncRNA-CIRPIL-p53 axis in the heart under reperfusion injury. The experiments were well designed and nicely executed. The conclusions are largely supported by the data. A few issues however remain as the following.

1. The authors need to carefully check the panels in Figure 1 since there are mis-representation in the main text. For example, Figure 1f is missing.
2. While apoptosis is a typical form of cell death of cardiomyocytes under I/R, the role of autophagy-related cell death is debatable. For clarity, I recommend removing the data about

autophagy from the manuscript, unless the authors can show strong evidence that autophagy induction here is responsible for cardiomyocyte death under I/R.

3. At what age did the authors measure cardiac function with echocardiography? How many days (weeks) after I/R? This question is pertinent to both overexpression and knockout mouse models. Along this line, other echocardiography parameters may be provided in the supplement. For example, heart rate, LVPW, etc.

4. The authors may consider change the order of supplemental figures. For example, now Figure S2 is described in the text earlier than Figure S1.

5. Did the authors identify other targets besides p53? A list of other potential targets may be provided.

6. Can the authors postulate a working model by which CIRPIL regulates the degradation of p53? For example, does the interaction between CIRPIL and p53 enhance E3 function?

7. It will be interesting to know whether supplementing the conserved lncRNA fragment in vitro as an oligo might protect cardiomyocytes against reperfusion damage.

8. The language requires significant polishing. There are too many grammatical errors, which severely affect the readability.

Reviewer #3 (Remarks to the Author):

In this manuscript, Yan et al. identified a down-regulated heart-enriched lncRNA-CIRPIL (Cardiac ischemia reperfusion associated p53 interacting lncRNA, IncCIRPIL) from the hearts of I/R mice. IncCIRPIL locates in the cytoplasm and physically interacts with p53, which leads to the cytoplasmic sequestration and the acceleration of subsequent ubiquitin-mediated degradation of p53. A human fragment of conserved IncCIRPIL sequence (Hcf-lncCIRPIL) mimicked the protective effects of the full-length IncCIRPIL on cultured human AC16 cells. Although this work contains both cellular and mouse genetic models to address the role of this lncRNA in I/R mice, it has some major defects in whole experimental design.

Major concerns:

1. Given that the mouse IncCIRPIL Tg and KO models have been generated, it is unclear why the mechanistic studies, especially the interaction with P53 in CMs, were not performed in vivo, but using cell cultures? Would crossing of IncCIRPIL Tg with KO mice would rescue all the phenotypes in I/R mice?

2. The method of cell line transfection and transfection efficiency were not fully clarified.

3. The full sequence of IncCIRPIL was unclear, the Chr5 location where IncCIRPIL locates has at least two ESTs, AA432627/461952, leading to a concern whether global KO of this lncRNA will affect other gene expressions.

4. Since this lncRNA shows such strong effect in P53 sequestration, why global KO of this gene has no visible phenotype during embryogenesis? P53 sequestration in KO heart was not included.

5. The human homolog has only 84% conservation from 17-71 nt with mouse gene, while the mouse gene has 381 bp. So the claim of conservation could not be fully supported.

Minor concerns:

1. RACE should be conducted to get the full sequence of IncCIRPIL.

2. Fig 1f should contain CM marker at least for NMCM. The RNA in situ results seems abnormal as it mainly overlaps with sarcomere in Sham adult CMs but evenly faded in the I/R CMs.

3. Fig.2c-e, data from three all shRNAs should be included. RNA FISH should be done to show OE and KD efficacy. Unbiased assay such as FACs should be included in quantification of apoptosis.

4. Fig3a, the OE effects vary a lot, seems like the mouse lines were not stable. RNA FISH should be included in heart sections. Fig3e, H&E and in situ apoptosis should be conducted. 3i, IHC should be included. How about fibrosis?

5. Fig. 4, how about the impact of this lncRNA in heart formation. P53 location /expression in the normal KO heart should be shown and compared with control.

6. Fig5, the smallest RNA motif interacting with P53 should be identified and mutated and overexpressed for functional validation. Gel shift is critical to be conducted to validate the RNA-protein interaction.

7. To confirm a lncRNA function, OE is not sufficient. Only loss-of-function could approve that a gene is indeed required for a certain bio-function. In this study, since this lncRNA has very high expression in CMs, further enhanced expression leads to a concern of over-dose-caused effect.
8. Genome-wide RNA-seq should be included in Tg and KO heart, focusing on the expression changes of p53 and its downstream genes.
9. Fig 7 should contain data from p53/lncRNA double knockdown, rather than double OE.
10. The evidence of human homology is vague. No evident of human lncRNA-P53 interaction.

Responses to Reviewers' Comments

Response to Reviewer #1:

Reviewer #1 (Remarks to the Author):

This study reported the identification of a new cardiac-enriched lncRNA, lncCIRPIL, and examined its role in protect heart against I/R injury. The underlying mechanisms were also elucidated. The study is well designed, and presented data strongly support the cardiac protection of lncCIRPIL. One concern is that previous studies have shown that inducing autophagy blunts I/R injury (10.1161/CIRCULATIONAHA.113.002416), but this contradictory was not sufficiently discussed. Other specific comments are listed below.

Reply: Thank you for the positive comments and the valuable suggestions. In this study, we only performed one simple experiment to test autophagy, which is not strength enough to support the role of lncCIRPIL in autophagy. In order to avoid any confusion and remain focused, we have decided removing the autophagy related data.

1. How is the expression of lncCIRPIL in I/R patients? This data will strongly increase the clinical relevance of the present study.

Reply: This is indeed a very important question. In fact, we are in the process of contacting colleagues from clinical institutions in China. However, largely due to pandemic restriction, this collaborative effort has been slowed. This certainly will be our next focus.

2. Introduction, "tumor suppressor p53 functions as a master 63 regulator of cell apoptosis and autophagy." Please add references here.

Reply: Thank you for the practical advice. We have added reference here. Please see reference "6" (Powell, E., Piwnica-Worms, D. & Piwnica-Worms, H. Contribution of p53 to metastasis. *Cancer Discov* 4, 405-414, doi:10.1158/2159-8290.CD-13-0136

(2014).).

3. **Figure 2, please also measure the activities of caspase-8 and caspase-9. In addition, activation of autophagy should be verified by accurately measuring autophagy flux using an autophagy inhibitor.**

Reply: Thank you for the insightful suggestion. To address your concern, we conducted additional experiments, and the new results provided evidences that LncCIRPIL inhibited apoptosis. LncCIRPIL overexpression reduced caspase-8 and caspase-9 activity and LncCIRPIL knockdown increased caspase-8 and caspase-9 activity *in vitro* (Figure i, which is incorporated into Figure 2e in the manuscript).

Figure i. Effects of LncCIRPIL on caspase-3, caspase-8, caspase-9 activity of NMCs subjected to A/R insult. n = 3. * $P < 0.05$ by one-way ANOVA followed by Tukey post hoc analysis.

We also detected caspase-8 and caspase-9 activity in other part of experiments as the supplementary evidences of apoptosis (Figure ii, which is incorporated into Supplementary Figure 2).

Figure ii. Effects of shRNA2 and shRNA3 of LncCIRPIL on Bcl2/Bax ratio in NMCs subjected to A/R insult. n = 3. *P < 0.05 by one-way ANOVA followed by Tukey post hoc analysis. e. Effects of shRNA2 and shRNA3 of LncCIRPIL on caspase-3, caspase-8, caspase-9 activity in NMCs subjected to A/R insult. n = 3. *P < 0.05 by one-way ANOVA followed by Tukey post hoc analysis.

Due to the preliminary and insufficient evidences of the effect of LncCIRPIL on autophagy, we abandoned this partial result.

4. Figure 3. The expression level of LncCIRPIL in other major organs of the LncCIRPIL-TG mice should be shown to confirm cardiac specificity.

Reply: Thank you for the suggestion. Accordingly, we performed another set of additional experiments measuring the expression of CIRPIL in major organs (heart, muscle, brain, kidney, liver, lung, spleen). The qRT-PCR data showed that LncCIRPIL expression was specifically up-regulated in the heart but not other tissues in LncCIRPIL TG mice (Figure iii, which is incorporated into Figure 3a in the manuscript).

Figure iii. LncCIRPIL expression in major organs of WT and LncCIRPIL-TG mice. n = 3 for each group. * $P < 0.05$ vs WT by 2-tailed Student t test.

5. Figure 6a. the quality of images were poor to support the statement that p53 level increased in nuclear of A/R+NC cells.

Reply: Thank you for the comment. We have repeated the staining and obtained better images to show the localization of p53. It showed that p53 translocated to the nucleus under the stimulation of H/R in NMCs (Figure iv, which is incorporated into Figure 6a in the manuscript).

Figure iv. Influence of LncCIRPIL overexpression on cytoplasmic and nuclear distribution of p53 in NMCs subjected to A/R insult by immunofluorescence staining. Scale bar = 20 μ m.

6. Figure 6b showed that LncCIRPIL reduced A/R-induced p53 protein over-expression in both cytosol and nuclear. But the data is not sufficient to support the conclusion that “In this study, LncCIRPIL promoted the ubiquitination of p53 by

enhancing its cytosol distribution” line 333.

Reply: Thank you for your excellent suggestions. We would delete the description. We would revise the description and also carefully check the description throughout the whole text.

7. The effect of lncRNA, such as LncRNA-CAIL in regulating p53 and autophagy and cardiomyocyte damage during I/R, and the mechanism by which P53 regulating autophagy and cell apoptosis has been established. The authors should further discuss the mechanistic novelty of the present study.

Reply: Thanks for this very important question. There is an excellent study illustrated that LncRNA-CAIL specifically inhibits autophagy by binding with p53 and blocking p53-mediated myocardin transcription. In our study, LncCIRPIL is a newly discovered cardiac protective factor by inhibiting apoptosis in myocardial ischemia reperfusion (I/R) injury and there was a strong and veritably binding between LncCIRPIL and p53 in pulldown results. On this basis, we found for the first time that lncRNA-mediated protein modification of p53, specifically ubiquitin degradation, and inhibited apoptosis. We stated the difference in the Discussion section (Page17, Line 339-344).

8. Why overexpression of lncCIRPIL increases ubiquitination of p53 after A/R injury but not in controls should be explained.

Reply: Thank you for the insightful question. In our results, no statistically significant changes in total protein levels of p53 were observed between NC and LncCIRPIL overexpressed group or WT and LncCIRPIL (TG) mice in normal circumstance. In normal circumstances, the expression level of p53 was low, and it was activated and transcribed under A/R or I/R stimulation¹. LncCIRPIL promoted ubiquitin degradation of p53 in protein level. Furthermore, LncCIRPIL was highly expressed in normal circumstances. Thus, the low level of p53 and the adequate expression of LncCIRPIL are the causes of no significant changes in total protein or ubiquitin degradation levels of p53 between NC and LncCIRPIL overexpressed group or WT and LncCIRPIL (TG) mice in normal circumstance.

- (1) Guedes, E. C. et al. High fat diet reduces the expression of miRNA-29b in heart and increases susceptibility of myocardium to ischemia/reperfusion injury. *J Cell Physiol* 234, 9399–9407, doi:10.1002/jcp.27624 (2019).

9. Previous studies showed that inducing cardiomyocyte autophagy blunt I/R injury (10.1161/CIRCULATIONAHA.113.002416), which is contract to the current findings. This discrepancy should be discussed.

Reply: Thank you for the insightful comment. In this study, we only performed one confirmative experiment to test autophagy, which, we agree, is not sufficient enough to fully support the role of lncCIRPIL in autophagy. Also we agree reviewer 2's comment (see below) on this point. In order to avoid any confusion and remain focused, we have removed the data related to autophagy.

Response to Reviewer 2

Reviewer #2 (Remarks to the Author):

In this manuscript, Jiang, Yang, and Zhang et al. identified a new lncRNA (CIRPIL) through microarray screening with cardiac ischemia/reperfusion (I/R) tissues. The authors went on to prove the role of CIRPIL as a beneficial player in cardiomyocytes under reperfusion injury in vitro. Both loss- and gain-of-function approaches have been employed. Furthermore, the authors generated transgenic and knockout mouse models to validate its role in vivo under cardiac I/R. Next, the authors identified a direct target of this lncRNA as p53. Mechanistically, lncRNA-CIRPIL stimulated the degradation of p53, thereby improving cardiomyocyte survival under I/R. Taken together, the authors carefully established a lncRNA-CIRPIL-p53 axis in the heart under reperfusion injury. The experiments were well designed and nicely executed. The conclusions are largely supported by the data. A few issues however remain as the following.

Reply: Thank you for the positive comments and valuable suggestions. We have performed additional experiments and try our best to revise the manuscript as suggested.

1. The authors need to carefully check the panels in Figure 1 since there are mis-

representation in the main text. For example, Figure 1f is missing.

Reply: Thank you for your patient advice. We apologize for the mistake. We have corrected the labeling of Figure 1. We also carefully checked and corrected all the figures.

2. While apoptosis is a typical form of cell death of cardiomyocytes under I/R, the role of autophagy-related cell death is debatable. For clarity, I recommend removing the data about autophagy from the manuscript, unless the authors can show strong evidence that autophagy induction here is responsible for cardiomyocyte death under I/R.

Reply: Thank you for the suggestion. We fully agree with your opinion. As the role of autophagy in myocardial I/R injury is still controversial and our data is quite preliminary, we have dropped the data and discussions related to autophagy.

3. At what age did the authors measure cardiac function with echocardiography? How many days (weeks) after I/R? This question is pertinent to both overexpression and knockout mouse models. Along this line, other echocardiography parameters may be provided in the supplement. For example, heart rate, LVPW, etc.

Reply: Thank you for the practical questions. We apologized for the missing of the information. In the present study, 8-10 weeks age and 25 ± 3 g weight mice were used in the *in vivo* experiments. We have added such a statement in the material method of “Establishment of cardiac ischemia/reperfusion injury in mice”. Furthermore, we have revised representative echocardiography by selecting an echocardiography which principal parameters closest to the mean in each group and supplemented markers of time and distance and parameters of LVIDd, LVIDs, LVEDV and LVESV in our echocardiography data (Figure v, which is incorporated into Figure 3b, 4b in the manuscript; Table i, which were incorporated into Table S3 in the Supplementary material).

Figure v. A. Representative cardiac echocardiography and statistical analysis of cardiac function of WT and LncCIRPIL transgenic (TG) mice subjected to cardiac I/R injury. **B.** Representative cardiac echocardiography and statistical analysis of cardiac function of WT and LncCIRPIL knockout (KO) mice subjected to cardiac I/R injury. EF, ejection fraction; FS, fractional shortening. $n = 9$ to 11 . $*P < 0.05$ by one-way ANOVA followed by Tukey post hoc analysis.

Table i Principal echocardiography parameters

Group	WT (n=9)	LncCIRPI N(TG) (n=9)	LncCIRPIL (TG) (n=9)	I/R+WT (n=9)	I/R+LncCI RPIL(TG) (n=9)	I/R+LncCI RPIL(KO) (n=9)
EF, %	71.91±0.70	72.55±0.89	71.74±0.89	52.57±0.58 ^a	61.17±1.07 ^b	42.61±1.60 ^b
FS, %	40.09±0.59	40.64±0.71	39.59±0.73	26.41±0.37 ^a	32.08±0.75 ^b	20.53±0.93 ^b
LVEDV, μ l	44.88±1.47	45.80±2.87	44.33±1.32	56.58±2.20 ^a	51.30±1.27	56.87±2.47
LVESV, μ l	12.59±0.48	12.67±1.05	12.56±0.61	26.82±1.06 ^a	19.87±0.59 ^b	32.55±4.16 ^b
LVIDd, mm	3.32±0.15	3.34±0.09	3.30±0.04	3.65±0.06 ^a	3.53±0.04	3.67±0.06
LVIDs, mm	1.99±0.03	1.98±0.07	1.99±0.04	2.69±0.04 ^a	2.38±0.03 ^b	2.91±0.05 ^b

^a $p < 0.05$ versus WT group; ^b $p < 0.05$ versus I/R+WT group.

They were analyzed by using two-way ANOVA, followed by Bonferroni's post hoc analysis. The data are expressed as means \pm SEM.

4. The authors may consider change the order of supplemental figures. For example, now Figure S2 is described in the text earlier than Figure S1.

Reply: Thank you for the careful advice. We apologize for the mistake. We have corrected the order of supplemental figures accordingly.

5. Did the authors identify other targets besides p53? A list of other potential targets may be provided.

Reply: Thank you for the suggestions. We did not test other targets besides p53. Based on MS data, the following interacting proteins (PRDX3, NDRG2, HSPA8, HSPA9) may be the potential targets of lncCIRPIL. We have included these proteins in Figure 5a. Please also see Page 9, Line177-180.

6. Can the authors postulate a working model by which CIRPIL regulates the degradation of p53? For example, does the interaction between CIRPIL and p53

enhance E3 function?

Reply: Thank you for the insightful comment. We think that CIRPIL may promote the degradation of p53 by enhancing E3 function. Our data showed that knockdown of CHIP, COP1 and MDM2 partially inhibited the degradation of p53 in the cytoplasm of NMCs under A/R insult (**Fig. 6f**), indicating that CIRPIL may enhance the activity of CHIP, COP1 and MDM2 directly or indirectly to facilitate the ubiquitination and degradation of p53.

7. It will be interesting to know whether supplementing the conserved lncRNA fragment in vitro as an oligo might protect cardiomyocytes against reperfusion damage.

Reply: Thank you for the good question. We planned and detected the specific function of hcf-LncCIRPIL plasmid and ASO in human AC16 cells, thus, we have only preliminarily demonstrated the role of conserved lncCIRPIL fragments in cardiomyocytes of mice under A/R stimulation by LDH release. Given the opportunity, we will further examine the specific role of the conserved lncCIRPIL fragment.

8. The language requires significant polishing. There are too many grammatical errors, which severely affect the readability.

Reply: We have checked the manuscript thoroughly and corrected the typos and language errors. We also invited an English-speaking scientist to help revise the language of our manuscript.

Reviewer #3:

Reviewer #3 (Remarks to the Author):

In this manuscript, Yan et al. identified a down-regulated heart-enriched lncRNA-CIRPIL (Cardiac ischemia reperfusion associated p53 interacting lncRNA, lncCIRPIL) from the hearts of I/R mice. LncCIRPIL locates in the cytoplasm and physically interacts with p53, which leads to the cytoplasmic sequestration and the acceleration of

subsequent ubiquitin-mediated degradation of p53. A human fragment of conserved lncCIRPIL sequence (Hcf-lncCIRPIL) mimicked the protective effects of the full-length lncCIRPIL on cultured human AC16 cells. Although this work contains both cellular and mouse genetic models to address the role of this lncRNA in I/R mice, it has some major defects in whole experimental design.

Reply: Thank you for the insightful comments. The comments are very important for us to improve the quality of the work. We have performed additional experiments to address these concerns.

Major concerns:

1. Given that the mouse lncCIRPIL Tg and KO models have been generated, it is unclear why the mechanistic studies, especially the interaction with P53 in CMs, were not performed in vivo, but using cell cultures? Would crossing of lncCIRPIL Tg with KO mice would rescue all the phenotypes in I/R mice?

Reply: Thank you for the insightful comment. Heart tissue is made up of many cells including cardiomyocytes, cardiac fibroblasts, endothelial cells, and vascular smooth muscle cells. Although lncCIRPIL is highly expressed in cardiomyocytes, p53 is an important transcription factor in a variety of cells and participated in many pathological mechanisms. To avoid the potential interference of other cell types, we isolated cardiomyocytes and investigated the specific mechanism of lncCIRPIL in cardiomyocytes during myocardial I/R injury. To supplement the issue, we further did a pulldown assay in heart tissues, and the result showed lncCIRPIL binds with p53 in hearts (Figure vi, which is incorporated into Supplementary Figure 7a).

Figure vi. Blotting of p53 pulled down by sense sequence of lncCIRPIL in heart tissue of mice. n = 3.

2. The method of cell line transfection and transfection efficiency were not fully clarified.

Reply: We apologized for missing of the information. In our study, NMCs and cell lines were transfected with a final concentration of 2.5 mg/L of plasmid vectors, shRNA, siRNA or ASO by lipofectamine 2000 reagent (Invitrogen, Carlsbad, America) in opti-DMEM circumstance, respectively. We have added such a statement in the material method of “cell transfection”. In Hek293T cell line, transfection efficiency of Flag-p53 or Flag truncations of p53 were shown in input group using Flag antibody (Figure 5f). The efficiency of hcf-LncCIRPIL plasmid and hcf-LncCIRPIL ASO in human AC16 cell line were shown in Figure 8A and Figure 8F.

3. The full sequence of lncCIRPIL was unclear, the Chr5 location where lncCIRPIL locates has at least two ESTs, AA432627/461952, leading to a concern whether global KO of this lncRNA will affect other gene expressions.

Reply: Thank you for the valuable question. We agree with your opinion that the region of Chr5 of lncCIRPIL locates has at least two ESTs. However, lncCIRPIL was described as starting from Chr5_111490828 and ending to Chr5_111492382 in GRCm38/mm10 (updated as Chr5_111638694-111640248 in GRCm39/m39) (Figure viA-B). To create a mouse Chr5_111490828-111492382 knockout specifically in C57BL/6 mice by CRISPR/Cas-mediated genome engineering, gRNA3 and gRNA4 were used to identify the sequence contains three exons (Exon 1, exon 2 and exon 3) of lncCIRPIL. The final sequence contains 2536 bases (Chr5_111490173-111492708 in GRCm38/mm10, updated as Chr5_111638039-111640574 in GRCm39/m39) was deleted successfully by CRISPR/Cas9 system (Figure viiC). To address the misgiving, we confirmed that the final sequence contained no other genes, except three predicted genes, which contained three exons same as lncCIRPIL (Figure viiD).

Figure vii. A. Sequence information of LncCIRPIL from noncode database (<http://www.noncode.org/>). B. Sequence information of LncCIRPIL from UCSC (<http://genome.ucsc.edu/>). C. Information of the final sequence deleted by CRISPR/Cas9 system from UCSC database (<http://genome.ucsc.edu/>). D. Information of the final sequence deleted by CRISPR/Cas9 system from NCBI (<https://www.ncbi.nlm.nih.gov/>).

4. Since this lncRNA shows such strong effect in P53 sequestration, why global KO of this gene has no visible phenotype during embryogenesis? P53 sequestration in KO heart was not included.

Reply: Thanks for the question. lncCIRPIL did not significantly change the expression of p53 under normal condition, while it significantly change p53 expression under ischemia condition (Figure viii, which is incorporated into Supplementary Figure 7a). We think this explains, at least partly, why we can obtain viable offsprings of CIRPIL KO mice.

Figure viii. The effects of lncCIRPIL knockdown on total, cytosol and nuclear levels of p53 protein in NMCMs subjected to A/R insult. n = 3. *P < 0.05 by one-way ANOVA followed by Tukey post hoc analysis.

5. The human homolog has only 84% conservation from 17-71 nt with mouse gene, while the mouse gene has 381 bp. So the claim of conservation could not be fully supported.

Reply: Thank you for the insightful comment. We have added the information in the result in order to be more accurate (Page 13, Line 263-266).

Minor concerns:

1. RACE should be conducted to get the full sequence of lncCIRPIL.

Reply: Thank you for the good suggestion. We have performed RACE experiment for several times. Unfortunately, we failed to obtain the full sequence of lncCIRPIL.

2. Fig 1f should contain CM maker at least for NMCM. The RNA in site results seems abnormal as it mainly overlaps with sarcomere in Sham adult CMs but evenly faded in the I/R CMs.

Reply: Thank you for the comment. To address this issue, we carried out the FISH and immunofluorescence staining assay to examine the location of lncCIRPIL. The data showed that lncCIRPIL was located in the cytoplasm and was not co-located with α -actin, the marker of cardiomyocyte either in NMCMs or adult cardiomyocytes (Figure ix, which is incorporated into Figure 1f).

Figure ix. Subcellular localization of lncCIRPIL in NMCs and adult mouse cardiomyocytes detected by Fluorescent in situ hybridization (FISH) & Immunofluorescence (IF) after A/R or I/R injury. n = 3. Scale bar=20 μ m. Red, lncCIRPIL, green, α -actin; blue, DAPI.

3. Fig.2c-e, data from three all shRNAs should be included. RNA FISH should be done to show OE ad KD efficacy. Unbiased assay such as FACs should be included in quantification of apoptosis.

Reply: Thank you for the suggestion. We further determined the function of the other two shRNAs (shRNA2, shRNA3) of lncCIRPIL on NMCs A/R injury. The results were shown in Figure x A-C (which is incorporated into Supplementary Figure 2b, d, e). The results showed that shRNA2 and shRNA3 of lncCIRPIL promoted apoptosis in NMCs under A/R stimulation, which were consistent with shRNA1.

Figure x. A. Effects of shRNA2 and shRNA3 of lncCIRPIL on LDH release from NMCs subjected to A/R insult. n = 3. * P < 0.05 by one-way ANOVA followed by Tukey post hoc analysis. **B.** Effects of shRNA2 and shRNA3 of lncCIRPIL on Bcl2/Bax ratio in NMCs subjected to A/R insult. n = 3. * P < 0.05 by one-way ANOVA followed by Tukey post hoc analysis. **C.** Effects of shRNA2 and shRNA3 of lncCIRPIL on caspase-3, caspase-8, caspase-9 activity in NMCs subjected to A/R insult. n = 3. * P < 0.05 by one-way ANOVA followed by Tukey post hoc analysis.

We performed FISH assay to determine the efficiency of LncCRIPIL plasmid, shRNAs of LncCRIPIL, and LncCRIPIL transgenic mice. The results were shown in Figure xi which is incorporated into Supplementary Figure 2a, 3b.

Figure xi. A-B. Transfection efficiency of LncCIRPIL overexpression plasmid and shRNAs in NCMCs and expression of LncCIRPIL in adult cardiomyocytes of LncCIRPIL transgenic mice by FISH & IF assay. Scale bar=20 μ m. Red, LncCIRPIL, green, α -actin; blue, DAPI.

We further performed experiments to determine the function of LncCRIPIL in A/R-induced cell death by flow cytometry for Annexin V- and PI-positive cells. The result shown that overexpression of LncCRIPIL decreased abundance of annexin V-positive apoptotic cells, whereas knockdown of LncCRIPIL did the opposite (Figure xii, which is incorporated into Supplementary Figure 2c).

Figure xii. Cell death by A/R was determined using flow cytometry to detect annexin V- and propidium iodide (PI)-positive cells. $n = 3$ for each group. $*P < 0.05$ by one-way ANOVA followed by Tukey post hoc analysis.

4. Fig3a, the OE effects vary a lot, seems like the mouse lines were not stable. RNA FISH should be included in heart sections. Fig3e, H&E and in situ apoptosis should be conducted. 3i, IHC should be included. How about fibrosis?

Reply: Thank you for the good suggestion. We examined the influence of lncCIRPIL overexpression on cardiomyocyte apoptosis in vivo. The data showed that TUNEL positive cells significantly decreased in LncCIRPIL(TG) mice after cardiac I/R injury (Figure xiii, which is incorporated into Figure 3h in the manuscript). In this study, we only explored the influence of LncCIRPIL on acute cardiac I/R injury. As fibrosis does not occur in this stage, we did not examine fibrosis related parameters. This is an interesting comment. We would investigate the influence of LncCIRPIL on cardiac fibrosis of chronic I/R model in the future.

Figure xiii. Effects of lncCIRPIL on apoptosis of heart tissue in mice subjected to I/R insult by TUNEL staining. n = 3. * $P < 0.05$ by one-way ANOVA followed by Tukey post hoc analysis; Scale bar=20 μ m.

5. Fig. 4, how about the impact of this lncRNA in heart formation. P53 location /expression in the normal KO heart should be shown and compared with control.

Reply: In our study, lncCIRPIL did not significantly change the expression of p53 under normal condition, while it significantly change p53 expression under ischemia condition (Figure xiv, which is incorporated into Supplementary Figure 7c). We obtained viable offsprings of CIRPIL KO mice and observed no morphological change in adult hearts. This may indicate that lncCIRPIL knockout did not change heart formation.

Figure xiv. The effects of lncCIRPIL knockdown on total, cytosol and nuclear levels of p53 protein in NCMs subjected to A/R insult. n = 3. * $P < 0.05$ by one-way ANOVA followed by Tukey post hoc analysis.

6. Fig5, the smallest RNA motif interacting with P53 should be identified and mutated and overexpressed for functional validation. Gel shift is critical to be conducted to validate the RNA-protein interaction.

Reply: Thank you for the excellent suggestions. We only found that the human homologous sequence region of LncCIRPIL is the specific region that binds to p53, and gain- or loss- of hcf-LncCIRPIL plays an important regulatory role in the apoptosis of AC16 cells under A/R treatment. It is a pity that we could not find more specific site, which is a limitation of our study.

7. To confirm a lncRNA function, OE is not sufficient. Only loss-of-function could approve that a gene is indeed required for a certain bio-function. In this study, since this lncRNA has very high expression in CMs, further enhanced expression leads to a concern of over-dose-caused effect.

Reply: Thank you for the insightful comment. LncCIRPIL knockout significantly increased the level of p53 and exacerbated cardiac I/R injury. We employed lncCIRPIL OE mice to correct its downregulation during cardiac I/R injury. Our results in transgenic mice demonstrated that under physiological conditions, overexpression of LncCIRPIL at high multiples had no adverse effects on the cardiac function, but showed a positive and protective effect against I/R injury (**Figure 3**).

8. Genome-wide RNA-seq should be included in Tg and KO heart, focusing on the expression changes of p53 and its downstream genes.

Reply: Thank you for the suggestion. We will certainly perform this experiment in the future.

9. Fig 7 should contain data from p53/lncRNA double knockdown, rather than double OE.

Reply: Thank you for the suggestion. P53-ko mice have broad biological effect, which led us to have concerns how double p53/lncRNA would help explain the specific function of lncRNA. Accordingly, we performed the double OE experiment, which we believed a better way to do and the data were shown in Figure xv, which were

incorporated into Figure 7e-g in the manuscript.

Figure xv. A. LDH release of NMCs co-transfected with lncCIRPIL siRNA and p53 siRNA. n = 3 for each group. * $P < 0.05$ by one-way ANOVA followed by Tukey post hoc analysis. B. Caspase-3, caspase-8, caspase-9 activity in NMCs co-transfected with lncCIRPIL siRNA and p53 siRNA. n = 3 for each group. * $P < 0.05$ by one-way ANOVA followed by Tukey post hoc analysis. C. BCL2/BAX ratio in NMCs subjected to A/R insult co-transfected with lncCIRPIL siRNA and p53 siRNA. n = 3. * $P < 0.05$ by one-way ANOVA followed by Tukey post hoc analysis.

10. The evidence of human homology is vague. No evident of human lncRNA-P53 interaction.

Reply: Thank you for the good suggestion. We further determined the interaction between hcf-LncCIRPIL and p53 in human AC16 cell line. The data showed that hcf-LncCIRPIL can bind to p53 in human AC16 cells (Figure xvi, which were incorporated into Figure 8a in the manuscript).

Figure xvi. Blotting of p53 pulled down by sense sequence of hcf-lncCIRPIL in human AC16 cells. n=3.

REVIEWERS' COMMENTS:

Reviewer #1 (Remarks to the Author):

The authors have adequately addressed my concerns.

Reviewer #2 (Remarks to the Author):

I want to thank the authors for carefully addressing my previous concerns.